# Comparison and optimization of cellular neighbor preference methods for quantitative tissue analysis

Chiara Schiller [1,2,3], Miguel A. Ibarra-Arellano [1], Kresimir Bestak [1,2,3], Jovan Tanevski [1,3] & Denis Schapiro [1,3,4,5] ✉

Studying the spatial distribution of cell types in tissues is essential for understanding their function in health and disease. A widely applied measure in spatial omics analysis is the pairwise neighbor preference of cell types, indicating whether two cell types frequently occur in close proximity to each other. While various neighbor preference methods exist, there is no clear guidance for selecting one over another. Here we present a comprehensive comparison of existing neighbor preference analysis methods. We systematically evaluate the method's underlying analytical steps and introduce COZI, a combination of analysis steps not previously described. We compare results from existing methods and COZI with respect to their ability to distinguish distinct tissue architectures and to recover neighbor preference directionality using two tissue simulations and two biological datasets. Overall, we delineate method-specific strengths and limitations and demonstrate that COZI provides sensitive and directional neighbor preference analysis for quantification of spatial data.

Understanding the spatial organization of tissues is crucial for identifying functional differences between conditions, such as healthy vs. diseased or respondent vs. non-respondent patients. In this context, spatial organization refers to the tissue architecture of cells and their interactions with each other. Cellular interactions and higher-order tissue architecture play a key role in determining tissue function, as they influence important processes such as immune responses and disease progression. With recent advances in spatial biology technologies[1–3], researchers can now capture these spatial characteristics with unprecedented detail on a cellular and molecular level. Alongside technological progress, there is also a diversity of computational methods to analyze the spatial features of the resulting datasets[4,5]. One of the most widely used measures is the pairwise neighbor preference (NEP) of two cell types within tissues[6,7]. These methods, broadly used in spatial proteomics and transcriptomics, are commonly referred to as cell-cell interaction[8,9], co-occurrence[10], neighborhood enrichment[11], or co-localization[12,13] analysis methods. We will use the term neighbor preference (NEP) for a directional understanding of an index cell and its neighbors without ligand-receptor information.

NEP methods have provided new biological insights in various areas including oncology, immunology, and developmental biology. E.g., distinct NEPs of immune and tumor cells across patients in triple-negative breast cancer hinted towards general differences in immune cell infiltration in tumors and led to their sub-classification into cold, hot, and mixed tumors[10]. In another study, antigen-presenting cells near the epithelium or fibroblasts were more predictive for progressors than non-progressors in patients with invasive breast cancer[14]. A study focusing on glioblastoma compared to brain metastasis revealed a difference in cancer cells interacting with the bordering

[1]Institute for Computational Biomedicine, Faculty of Medicine, Heidelberg University and Heidelberg University Hospital, Heidelberg, Germany. [2]Faculty of Biosciences, Heidelberg University, Heidelberg, Germany. [3]Translational Spatial Profiling Center (TSPC), Heidelberg, Germany. [4]Institute of Pathology, Faculty of Medicine, Heidelberg University and Heidelberg University Hospital, Heidelberg, Germany. [5]IFOM ETS - the AIRC Institute of Molecular Oncology, Milan, Italy. ✉e-mail: denis.schapiro@uni-heidelberg.de

brain parenchyma[15]. Despite the power of these analyses, the methods used to infer NEPs vary greatly, with each approach offering distinct advantages and limitations.

Tissues show subtle, heterogeneous spatial differences that can lead to markedly different results depending on the NEP method used. One key difference between methods is how they capture NEP directionality. Some studies assume symmetry in their analysis, meaning that any given two cell types mutually prefer to be close to each other to the same extent[14,16]. Other studies provide NEP scores for two cell types in both directions, aiming to capture the directionality of NEPs[8,10,11]. Directional NEP can indicate patterns of communication or migration. For example, the spatial gradient of immune cells migrating from the stroma to tumor cores may indicate immune activation or evasion[17,18]. In wound healing, the state of infiltration of immune cells and their interaction with myofibroblasts corresponds to normal healing, cold or hot fibrosis[19,20]. Understanding and quantifying changes in these asymmetric spatial signatures is essential for comprehending tissue dynamics and developing therapeutic targeting strategies.

Neighbor preference methods are widely applied in spatial proteomics and transcriptomics. Numerous computational approaches exist but differ in their underlying concepts and their corresponding results. To guide method selection, we systematically evaluate commonly used NEP methods. We evaluate their performance on simulated and real-world datasets ranging from the detection of cohort differences to the identification of more subtle and directional NEPs. We focused on the following methods or analysis toolboxes providing functions for calculating NEP scores: histoCAT[8], Spatial Enrichment Analysis (SEA)[10], Giotto[16], IMCRtools (classic)[21], Squidpy[22], Scimap[23] and Misty[9]. We also included CellCharter[11], which provides a function for NEP of niches that can also be applied to cell phenotypes. While being frequently used, there is no comparison or guide of these conceptually similar NEP methods and their underlying analysis steps to identify frequently occurring NEPs. Here, we deconstruct the methods into their underlying analysis steps and evaluate their performances on simulated ground-truth and biological tissue data. We identify the ideal analysis steps for (1) their discriminatory power to distinguish different tissue architectures and (2) their ability to recover the directionality of NEPs.

Finally, we propose an alternative NEP analysis approach as a combination of analysis steps not previously described. We identified that a conditional z-score (COZI) enables both the sensitive detection of differences in NEPs and their directionality. We demonstrate the biological utility and limitations of the different methods by applying them to two simulated datasets, a triple negative breast cancer and a myocardial infarction dataset, both studying immune cell infiltration. We thereupon introduce the conditional cell ratio (CCR) as additional score helping to interpret conditional NEP scores. Our findings highlight the heterogeneity of NEP method results, the advantages of combining the best analysis features from existing methods and offer a comprehensive guide for researchers and method developers in spatial omics analysis. By defining a common vocabulary, performing a systematic evaluation, and introducing an innovative approach, we aim to enhance the accuracy and interpretability of spatial analyses.

## Results

### Unifying concepts and algorithms in NEP analysis

To systematically compare existing pairwise neighbor preference (NEP) methods, we first identified their shared concept, algorithmic components, key differences, and the terminology used to describe them. Conceptually, NEP methods ask: "What does a cell of type A 'see' in its local environment?". For an index cell (e.g., an immune or tumor cell), NEP characterizes the composition of its direct neighbors and assesses whether the observed pattern of neighbors deviates from what would be expected under a spatially randomized null model. At the tissue level, NEP summarizes the extent of local spatial organization or avoidance between cell types, reflecting biological processes such as attraction, repulsion, or compartmentalization on a local scale. Importantly, NEP is not limited to distinguishing between random and structured configurations. It is equally suited to differentiating weak from strong interaction patterns and to capturing how interaction strengths vary across biological cohorts, such as between health and disease. In this way, NEP provides a window into how cells are functionally organized within the microenvironment, beyond mere abundance or proximity. At the same time, NEP does not capture higher-order spatial structures, such as global tissue architecture, multicellular motifs, or long-range interactions beyond immediate neighbors. In this sense, NEP is not only a descriptive statistic but also a hypothesis-testing framework: it enables testing whether local interactions between cell types are non-random and potentially biologically meaningful.

While NEP methods are widely applied in spatial biology, their underlying computational steps vary, making direct comparisons challenging. While some methods are toolboxes with more specific function names, we consistently refer to the underlying NEP method by their package name throughout the manuscript (Methods). By deconstructing these NEP methods into their core analysis steps, we established a unified framework that enables a structured evaluation of their performance (Fig. 1a–d). We identified three fundamental steps common to all NEP methods: (1) neighborhood definition, (2) neighbor quantification, and (3) NEP scoring (Fig. 1a–c). These steps and their combinations determine how spatial relationships between cell types are measured and how biological data can be interpreted. In the following, we describe each step and outline the algorithmic variations across methods.

**Neighborhood definition.** All methods first define a cellular neighborhood, the space around an index cell to determine its neighbors (Fig. 1a). This is based on either (i) a fixed distance (e.g., Euclidean in pixels/μm) or (ii) a graph-based approach (e.g., Delaunay triangulation, k-nearest neighbors). Most methods use a single neighborhood, except MISTy, which incorporates multiple distinct spatial views. We harmonized neighborhood definitions across methods for comparability of results.

**Quantification.** After defining the cellular neighborhood, the next step is quantifying neighboring cells within the specified space (Fig. 1b). This can be done in two ways: (i) Non-directional counting, which aggregates neighbors (e.g., red-purple) without considering the index cell's identity, and (ii) bi-directional counting, which assigns separate counts for each direction (e.g., "red-purple" vs. "purple-red"). The latter can capture asymmetries in cell-cell preferences, revealing directional preference of one cell type to another. Most methods normalize neighbor counts using either "total" normalization, dividing by the total number of index cells, or "conditional" normalization, considering only index cells with at least one neighbor of the specified neighbor cell type. One exception: CellCharter does not normalize by the number of cells but rather the fraction of edges in the connectivity graph of an image.

**NEP score.** The final step is calculating a NEP score which is a unique score for each method (Fig. 1c). E.g., methods can either directly provide the scaled and normalized neighbor counts as NEP score, or assess and report whether a cell type pair is neighboring more often than expected by chance through permutation testing. Common statistical scores for NEP include z-scores, p-values and normalized counts. CellCharter reports the difference between the observed and the expected number of links derived from node degrees in the network, which is computationally faster than permutation testing[11]. MISTy treats NEPs as an inductive task, providing values for the variance

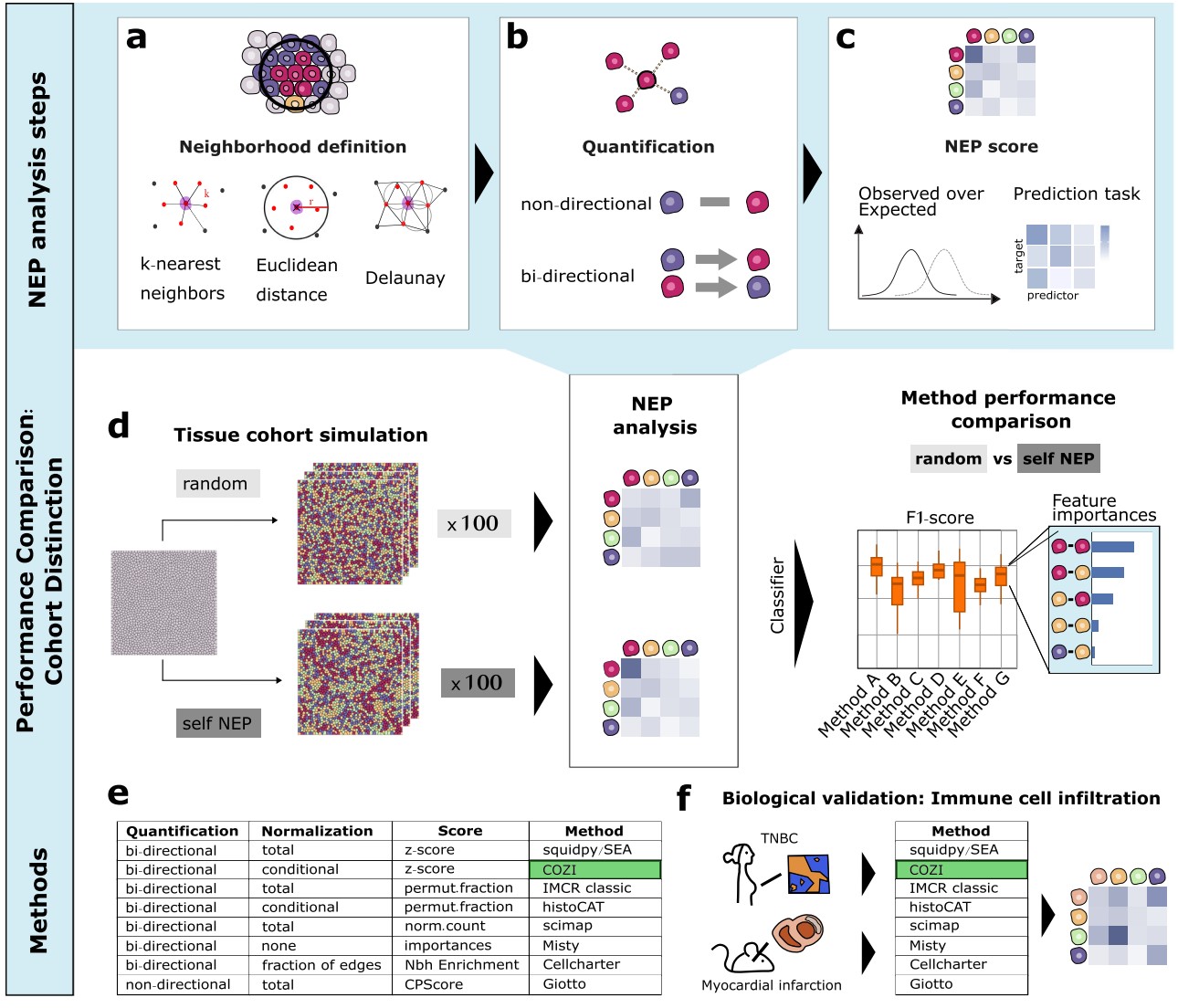

**Fig. 1 | Overview of neighbor preference (NEP) method performance comparison.** Three common NEP analysis steps were identified. **a** Neighborhood definition: Graph (Delaunay triangulation or k-nearest neighbors (kNN)) or distance based (Euclidean) neighborhood definition. User-provided parameters for Euclidean radius (r) or kNN (k) in red. **b** Quantification: Neighboring cells (within neighborhoods) are counted non- or bi-directionally. Non-directional counting aggregates neighbors (e.g., red-purple) without considering the index cell's identity. Bi-directional counting assigns separate counts for each direction (e.g., "red-purple" vs. "purple-red"). **c** NEP score: NEP scores are determined. (Normalized) Neighbor counts are either assessed if they occur more often than expected or a prediction task is formulated for predictor-target cell types. **d** Schematic of the experimental design. Different NEP cohorts were designed and analyzed using the methods. A random forest classifier was trained with the NEP scores to evaluate how well the cohorts could be distinguished examining the F1 scores and feature importances. **e** Overview of methods and their underlying algorithmic steps to provide NEP scores. **f** Biological validation of the methods with a triple negative breast cancer (TNBC) dataset and a myocardial infarction dataset studying immune cell infiltration.

explained[9]. Overall, all methods infer pairwise NEP scores, either non- or bidirectional.

Building on this systematic deconstruction of NEP analysis steps, we introduce the conditional z-score (COZI) as an alternative approach not yet implemented by existing methods. Our overview of analysis steps showed that various analysis step combinations were implemented by known methods (Fig. 1e). However, one combination involving bi-directional counting and statistical scoring was missing: "conditional" normalization with a z-score, which no current method incorporates. To fill this gap, we developed and implemented COZI. In the following, we compare the performance of all existing methods and our in-house developed COZI (Fig. 1e) on simulated and biological data (Fig. 1f) and examine how differences in analysis steps explain performance variations and biological interpretability.

## Assessing cohort distinction sensitivity of NEP methods

We initially evaluated the methods by assessing their ability to distinguish tissue cohorts based on the spatial signatures they capture (Fig. 1d). A key goal in spatial biology is identifying distinctions between biological cohorts (e.g., healthy vs. diseased, responders vs. non-responders) based on tissue architecture. NEP analysis generates one preference score per image for each pair of cell types. This allows comparisons such as whether the purple cell type exhibits a stronger preference to neighbor the red cell type in one image than in another. When multiple samples are available per cohort, these per-image NEP scores can then be compared across samples to detect cohort-level differences, a common practice in the field. Our first experimental setup is built on this common practice: If a distinct NEP signature exists in one cohort but not the other, the methods should detect this

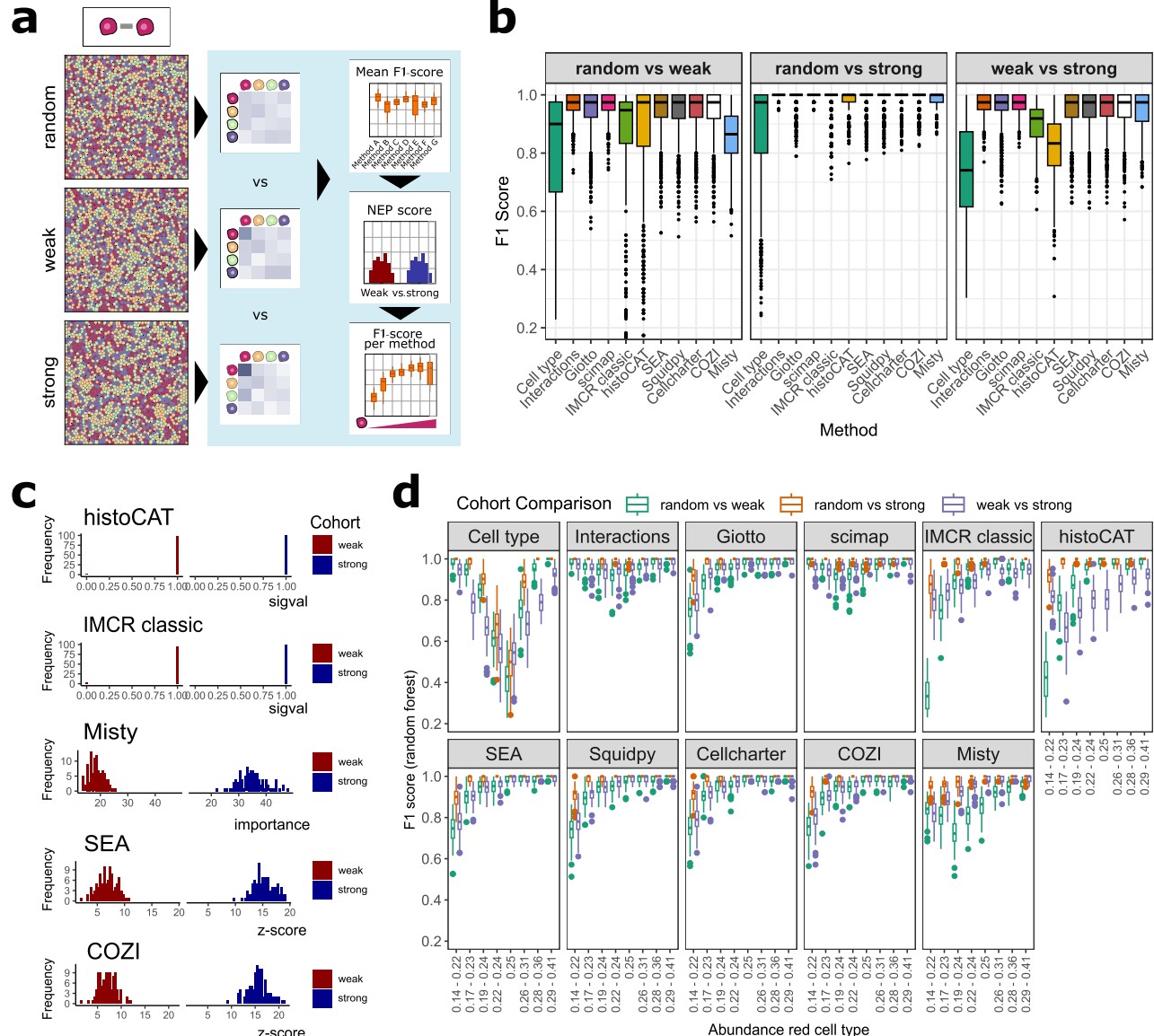

**Fig. 2 | Systematic comparison of cohort distinction ability based on neighbor preference (NEP) analysis results with a random forest classifier. a** Exemplary in silico tissue (IST) images for the three adjacency cohorts: random, weak, and strong self-preference of the red cell type. The red cell type was abundant at 25% in all depicted images. Schematic workflow: Cohorts were analysed by all NEP methods and binary classification was evaluated by assessing mean F1 scores (**b**), the true NEP scores (**c**) and the F1 scores across increasing abundance of the red (self-preference) cell type (**d**). **b** The boxplots show the mean of F1-scores per cohort distinction task across red cell type abundance groups. Cohort comparisons were made for random vs. weak, random vs. strong, and weak vs. strong self-preference. 100 images per cohort were simulated and analyzed. Data are presented as box plots showing the median (center line), interquartile range (box; 25th–75th percentiles), and whiskers extending to 1.5 × IQR. Outliers are shown as individual points. **c** NEP score distribution of red cell self interaction NEP scores of weak and strong self-preference cohorts from histoCAT, IMCRclassic, Misty, SEA, and COZI. All cell types were equally abundant at 25%. 100 images per cohort were simulated and analyzed. **d** The same experiment as in (**b**): F1 scores plotted per method across increasing cell type abundance cohorts of the red cell type. Data are presented as box plots showing the median (center line), interquartile range (box; 25th–75th percentiles), and whiskers extending to 1.5 × IQR. Outliers are shown as individual points. Source data are provided as a Source Data file.

difference. Therefore, instead of assessing how well methods reconstructed the underlying tissue structure, we initially focused on their ability to differentiate between the cohorts. Since real datasets lack ground truth spatial architecture, we used the in silico tissue (IST) generation framework[24] to simulate tissue cohorts with known differences in NEP (Fig. 1d). The simulation, which represents a simplified version of tissue architecture, uses a linear cell-cell adjacency matrix with one NEP score per cell type pair as input. The method therefore further allows us to directly compare simulation input and NEP output. To evaluate if the methods could capture the simulated cohort differences, we trained classifiers with each method's NEP scores and

evaluated the cohort classification performance by assessing F1-scores and feature importances (Fig. 1d). In the following, we systematically identify and explain performance differences based on the outlined NEP analysis steps.

## The importance of continuous scores for detection of NEP differences

First, we simulated tissue cohorts for cohort distinction tasks, where the red cell type displayed varying levels of self-preference, ranging from random distribution to weak and strong self-preference (Fig. 2a). These different levels of self-preference were simulated across various

abundances of the red cell type to evaluate the sensitivity of the methods for detecting NEPs of both low- and high-abundance cell types. We performed binary classifications comparing random vs. weak, random vs. strong, and weak vs. strong self-preference within abundance groups, ensuring that the comparisons focused on spatial patterns rather than differences in abundance. In the following analysis, we examine overall F1 scores, raw NEP scores, and F1 scores across increasing red cell type abundance, in order to assess general method performance and explain the observed performance differences (Fig. 2a).

All methods effectively distinguished the simulated tissue cohorts based on NEP differences, performing better than a random model and exceeding the classification based solely on cell type abundance differences (Fig. 2b, and Supplementary Fig. 1a). The classification between the random and strong self-preference cohorts was nearly perfect across all methods (F1-score -1, Fig. 2b, second panel), as these cohorts exhibited the most pronounced spatial differences (Fig. 2a). Distinguishing between random and weak cohorts, and between the weak and strong cohorts resulted in slightly lower but still high performance (F1-score > 0.8, Fig. 2b, first and third panels). Of note, cohort classification based on cell type abundance differences alone also achieved F1-scores above 0.5 (random classification) due to inherent limitations in IST generation (Supplementary Fig. 2a and b). In this experimental setup, all methods performed very well and the only notable performance decline was observed for IMCRtools classic and histoCAT, particularly in the weak vs. strong cohort classification (Fig. 2c, third panel).

The performance drop of IMCRclassic and histoCAT can be attributed to their use of the significance value (sigval) metric for computing NEP scores[8]. We examined the raw NEP scores of the red cell type to itself in weak and strong IST cohorts, each with 25% red cell type abundance (Fig. 2c). In both histoCAT and IMCRclassic, sigval scores were largely 1, making it impossible to distinguish the red self-preference between the weak and strong cohorts. Both methods follow a simple question, assessing whether the underlying NEPs are significantly different from a random NEP. Therefore both methods provide a categorical response to a two-tailed test, with NEP scores of −1, 0, and 1. In cases of strong NEP, the score is 1 in most images, as nearly all randomly shuffled images have fewer neighbors than the original. In contrast, other methods like Misty (feature importances), SEA or COZI (z-scores) produce normally distributed NEP scores, allowing for clearer differentiation between weak and strong cohorts (Fig. 2c, and Supplementary Fig. 3). NEP scores across all cell-cell interactions in the random, weak and strong cohort highlight only Scimap to not have normally distributed scores. Scimap scales NEP scores between −1 and 1, leading to skewed NEP scores even in the randomly organized tissue (Supplementary Fig. 3g).

We further evaluated the method performances across different red cell type abundance levels. The performance drop of IMCRclassic and histoCAT became more pronounced as the red cell type abundance increased (Fig. 2d). With higher abundance, methods with categorical NEP scores struggled to characterize subtle NEP differences, whereas methods with continuous scores performed better. Across different cell type abundance groups, Misty and Scimap exhibited F1 score trends similar to those driven by cell type abundance differences (Fig. 2d, first panel) or simple pairwise neighbor counts (Fig. 2d, second panel). Cohorts with the highest and lowest red cell type abundances were the easiest to distinguish based on simple cell type or neighbor counting (F1 - 1). Misty and Scimap followed this trend, performing best when abundance differences were largest. In contrast, methods that use permutation testing or report differences between observed and expected NEPs performed worst at low red cell type abundances and best at high abundances with self-preference. This was expected, as NEPs of lowly abundant cell types are harder to capture compared to those of highly abundant cell types. Since Misty

and Scimap do not compare observed to random NEPs, their NEP scores are more strongly influenced by cell type abundance differences. We will revisit this observation in our second simulation experiment.

## Unlocking NEP directionality with conditional count averaging

After evaluating the methods' sensitivity to NEP differences, we investigated whether those providing bi-directional neighbor quantification (Fig. 1b) could capture NEP directionality using a simulated dataset. Here, directionality refers to one cell type showing a stronger preference for another as a neighbor than vice versa, for instance, immune cells infiltrating a tumor prefer tumor cells as neighbors more than the reverse. Therefore, we simulated a dataset with asymmetric cell-cell adjacencies, where the red cell type had a stronger preference for the yellow cell type than the other way around (Fig. 3a). We generated three levels of cross-preference random, weak, and strong— while varying red cell type abundance. Although simulated and observed cell type abundances differed, we ensured cohorts included both higher red/lower yellow and lower red/higher yellow cell type abundances (Supplementary Fig. 2b). In this experiment, we added CellCharter's package option to not include homotypic interactions, which we will call CellCharter* in the following.

We first assessed cohort distinction performance using F1 scores. We found that all methods outperformed the cell type abundance and random label shuffling baselines, except for distinguishing random from weak self-preference, where the cell type abundance outperformed some methods (Supplementary Fig. 1b, Supplementary Fig. 4a). Across all methods, F1 scores increased with higher red cell type abundance (Fig. 3b). Misty and Scimap, as well as CellCharter*, followed similar F1-score patterns to the cell type abundance or interactions counts. For Giotto, we duplicated outputs to mimic bi-directional neighbor counting, enabling comparison of non-directional to bi-directional quantification. Giotto performed similarly well to SEA, COZI, CellCharter, and Squidpy in the cohort classification task, methods that inherently use bi-directional neighbor quantification (Supplementary Fig. 4a, and Fig. 3b). Investigating the feature importances of the methods in the 25% abundance random vs. strong cohorts showed that only Misty, COZI and histoCAT had notably high feature importances for the red-yellow cross-preference (Supplementary Fig. 4b). CellCharter and Scimap showed high feature importances for the other direction, yellow-red, while none of the other methods captured a difference (Supplementary Fig. 4b). This led us to further investigate how and if these methods capture differences in directional NEP pairs.

While F1 scores measure overall classification performance in distinguishing different cohorts, cosine similarity specifically quantifies how well the methods recover the true adjacency patterns by comparing the methods-derived NEP scores to ground-truth cell-cell adjacencies. This distinction is crucial, as a method may achieve high classification performance without necessarily capturing the correct underlying spatial relationships. Therefore, we evaluated the recovery of differences in the ground truth cell-cell adjacencies by measuring cosine similarities between the model's feature importances and ground-truth cell-cell adjacency values (Fig. 3c). Higher cosine similarity indicates better alignment with the true adjacency structure. A key difference emerged between neighbor count averaging methods: The bi-directional methods SEA, IMCRtools classic, CellCharter and Squidpy yielded cosine similarities comparable to Giotto, the only non-directional method (Fig. 3c). The three methods—SEA, IMCRtools classic, and Squidpy—normalize neighbor counts by the total number of cells of the same type as the index cell. Their raw NEP scores for red-yellow and yellow-red were indistinguishable, confirming that the methods do not capture NEP directionality (Fig. 3d). In contrast, histoCAT and COZI showed higher cosine similarities than the named bi-directional methods (Fig. 3c). Both perform conditional normalization,

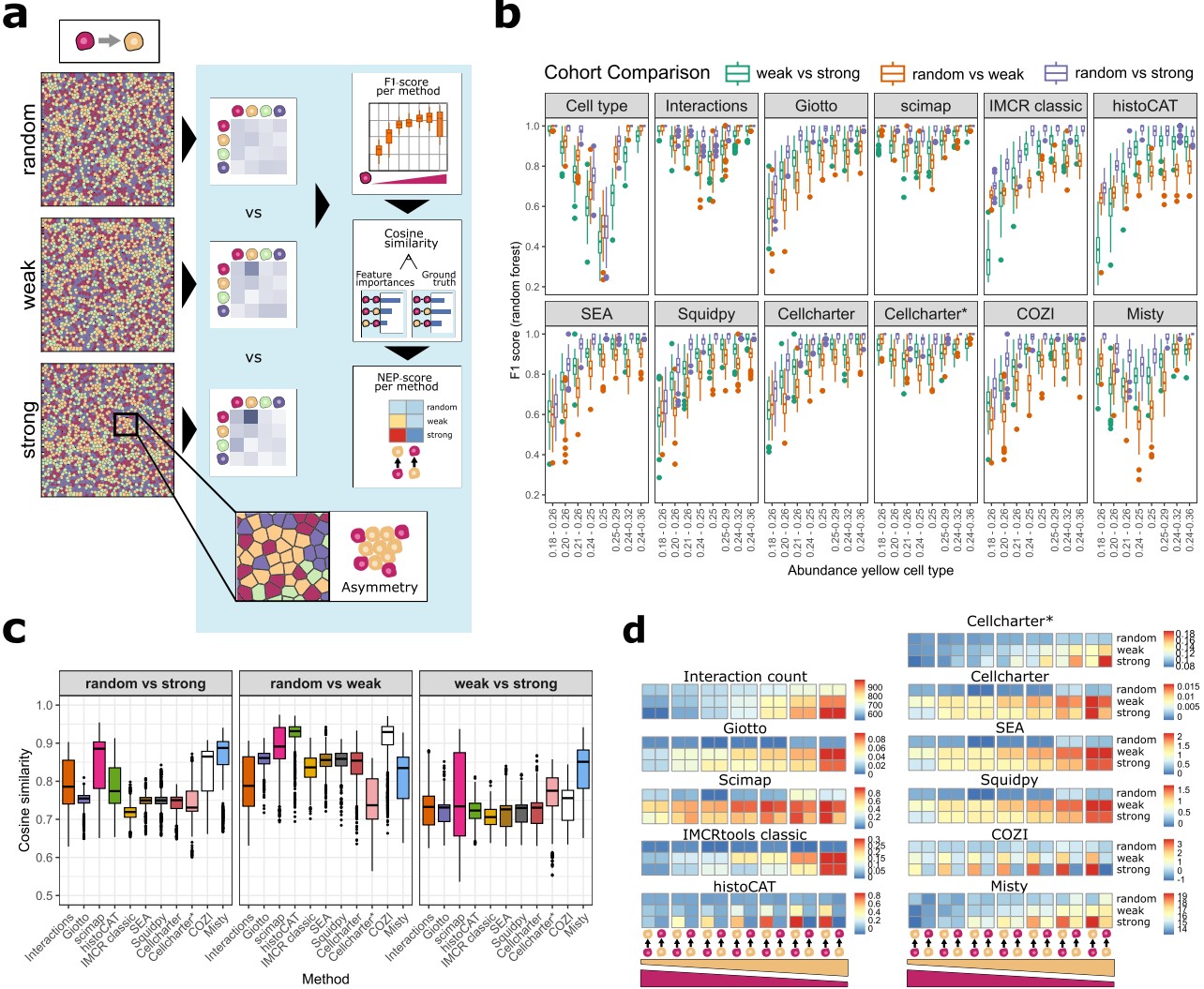

**Fig. 3 | Systematic comparison of the ability to recover neighbor preference (NEP) directionality. a** Exemplary IST images for the three adjacency cohorts - random, weak, and strong - with a red-to-yellow cell type cross-preference. The red cell type was abundant at 25% in all three depicted images. **a** Schematic depiction of (**b**–**d**). **b** F1 scores of cohort classification tasks per method across cohort comparisons. 100 images per cohort were simulated and analyzed. Data are presented as box plots showing the median (center line), interquartile range (box; 25th–75th percentiles), and whiskers extending to 1.5 × IQR. Outliers are shown as individual points. **c** Mean cosine similarities between feature importances and ground truth adjacency value differences between cohorts were calculated per method. Boxplots show mean cosine similarities per cohort distinction task.

Cohort comparisons between random vs. weak, random vs. strong and weak vs. strong cross-preference. 100 images per cohort were simulated and analyzed. Data are presented as box plots showing the median (center line), interquartile range (box; 25th–75th percentiles), and whiskers extending to 1.5 × IQR. Outliers are shown as individual points. **d** Raw NEP scores per method for the random, weak, and strong cohort for the red-yellow and the yellow-red NEP. Heatmap of inter-action counts and NEP scores for Giotto, Scimap, IMCRtools classic, HistoCAT, CellCharter* (without homotypic interactions), CellCharter, SEA, Squidpy, COZI, and Misty. Red cell type abundances range from higher than the yellow cell type to lower than the yellow cell type from left to right. Color legends indicate the method specific NEP score. Source data are provided as a Source Data file.

normalizing index-neighbor counts by the number of cells of the index cell type with at least one link to a neighbor cell type. This approach leads to the recovery of the simulated red-yellow NEP but not vice versa consistently across abundance levels (Fig. 3d). Both versions of CellCharter also capture directionality, as stated in the method. CellCharter including homotypic interactions captures a correct directional red-yellow preference in cohorts with higher abundance of the yellow cell type, while it does not capture directionality in equal or low red abundance cohorts (Fig. 3d). Without permutation testing in CellCharter, there is a potential cell type abundance bias. The total number of cells of the red type is proportional to the total number of edges which decreases the ability to capture the preference of a highly to a lowly abundant cell type. CellCharter* excluding homotypic edges in the network consistently captures NEP differences between the two

directions across cell type abundance differences, but the other way around than simulated (Fig. 3d).

Misty and Scimap also reached high F1-scores (Supplementary Fig. 4a), cosine similarities (Fig. 3c), and differences between the red-yellow and yellow-red neighbor preferences (Fig. 3d). Both methods, however, were also affected by cell type abundances throughout this study (Fig. 3b). Both methods falsely recover a higher score for the yellow-red than for the red-yellow NEP for low abundances of the yellow cell type (Fig. 3d). This observation is explainable by their score: Scimap does not provide a z-score, but the scaled neighbor count of the red-yellow cell co-occurrences divided by the number of red cells. A low number of yellow cells inflates the yellow-red score compared to red-yellow. Similarly, Misty provides values of the variance explained based on a prediction task not corrected by permutation testing. As

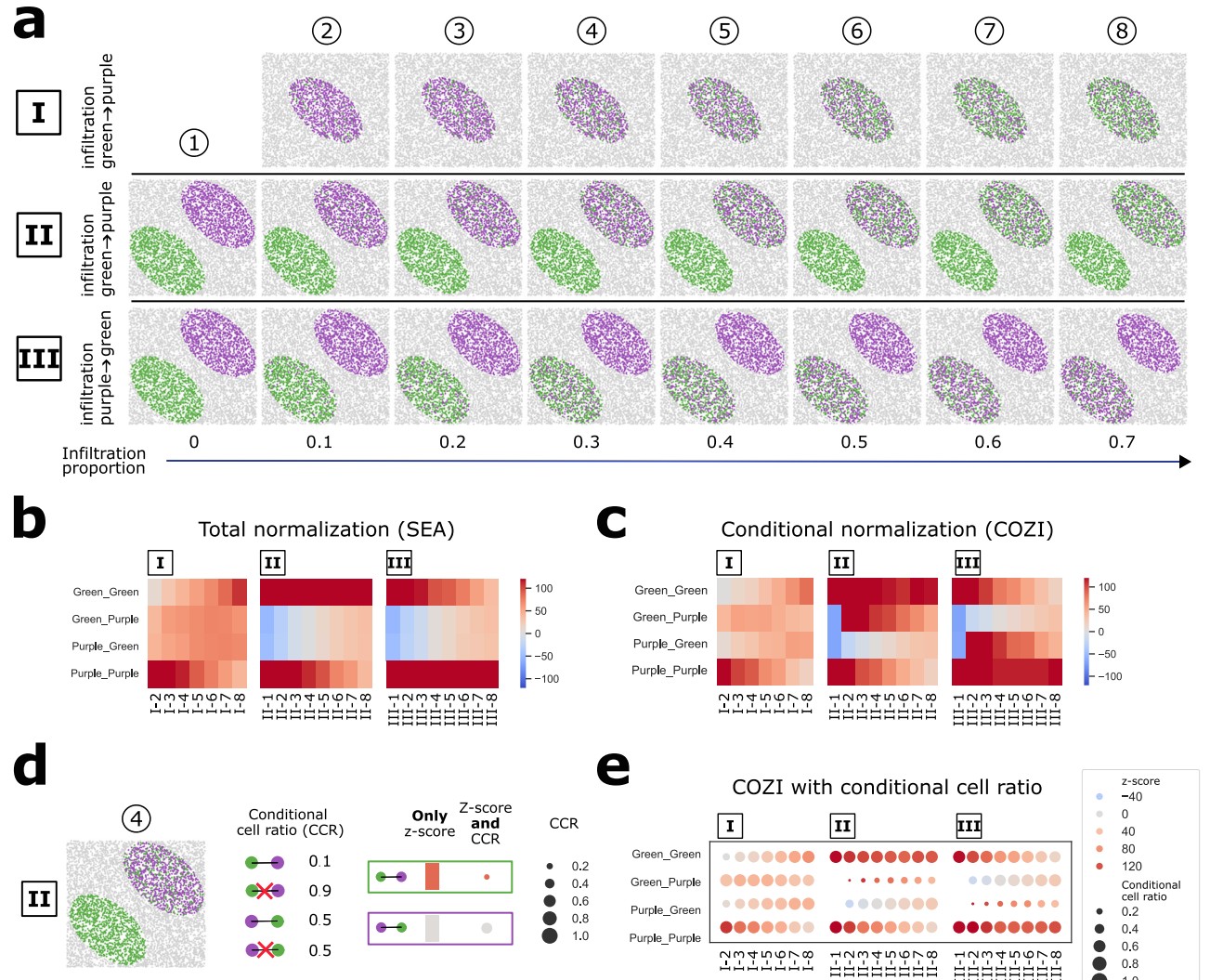

**Fig. 4 | SpaSim simulations to study directionality of neighbor preference (NEP) scores in cell infiltration patterns. a** SpaSim simulation of three datasets: (I) one purple cluster infiltrated by green cells, (II) a purple cluster infiltrated by green cells in the presence of a green cluster, and (III) a green cluster infiltrated by purple cells in the presence of a purple cluster. *n* = 1 image was generated and analysed per infiltration and cohort. **b**, **c** NEP analysis using (**b**) SEA (z-score, total normalization) and (**c**) COZI (z-score and conditional normalization) on simulated datasets I-III. Color legends indicate the method specific z-score. **d** Schematic explanation of result interpretation using z-scores with and without conditional count ratio (CCR). **e** COZI z-scores as in (**c**) but with dot size indicating conditional cell ratio. Source data are provided as a Source Data file.

Scimap and Misty do not provide scores based on permutation testing or expected neighbor counts, both do not account for differences in cell type abundances—in comparison to how a permutation test does. Therefore, it is questionable how well they can correctly recover NEPs of a highly abundant cell type to a lowly abundant one.

Overall, conditional averaging was the decisive algorithmic step in recovering NEP directionality in our simulation experiments, making it a key feature in the discussed methods. The combination of conditional count averaging and z-score calculation performed best across the designed IST simulation experiments. Due to its limitation to distinguish similar NEPs with varying degrees of strength, histoCAT's distinction ability drops when comparing F1 scores of the weak vs the random cross-preference cohort, thus confirming our previous results (Supplementary Fig. 4a). The COZI performance stays as high as the other permutation-based methods and recovers directionality. Giotto, IMCRtools classic, SEA, and Squidpy do not recover directionality due to total count normalization. CellCharter and CellCharter* do capture directionality but either not across all cell type abundance groups or in the opposite direction. Scimap and Misty are primarily affected by cell type abundance differences.

## Studying directional infiltration patterns with the conditional cell ratio (CCR)

To investigate how directionality can inform the interpretation of infiltration patterns on a single-sample level, we next used SpaSim[12], a simulation framework that generates cell clusters infiltrated by other cell types, resembling scenarios such as tumor–microenvironment interactions. Unlike the linear adjacency setup in the IST framework, SpaSim produces spatial arrangements that more closely mimic biological tissue. We simulated three infiltration scenarios with purple and green cells (Fig. 4a): (I) one purple cluster infiltrated by increasing numbers of green cells, (II) a purple cluster infiltrated by green cells in the presence of a green cluster, and (III) the reverse scenario, with a green cluster infiltrated by purple cells.

When applying NEP analysis, total normalization (SEA) consistently produced global, direction-agnostic scores: both green–purple and purple–green interactions were scored similarly, and only broad trends such as overall preference or avoidance were detected (Fig. 4b). By contrast, conditional normalization (COZI) distinguishes directionality between cell types. For example, in scenario I, COZI captured a decreasing green–purple preference alongside an

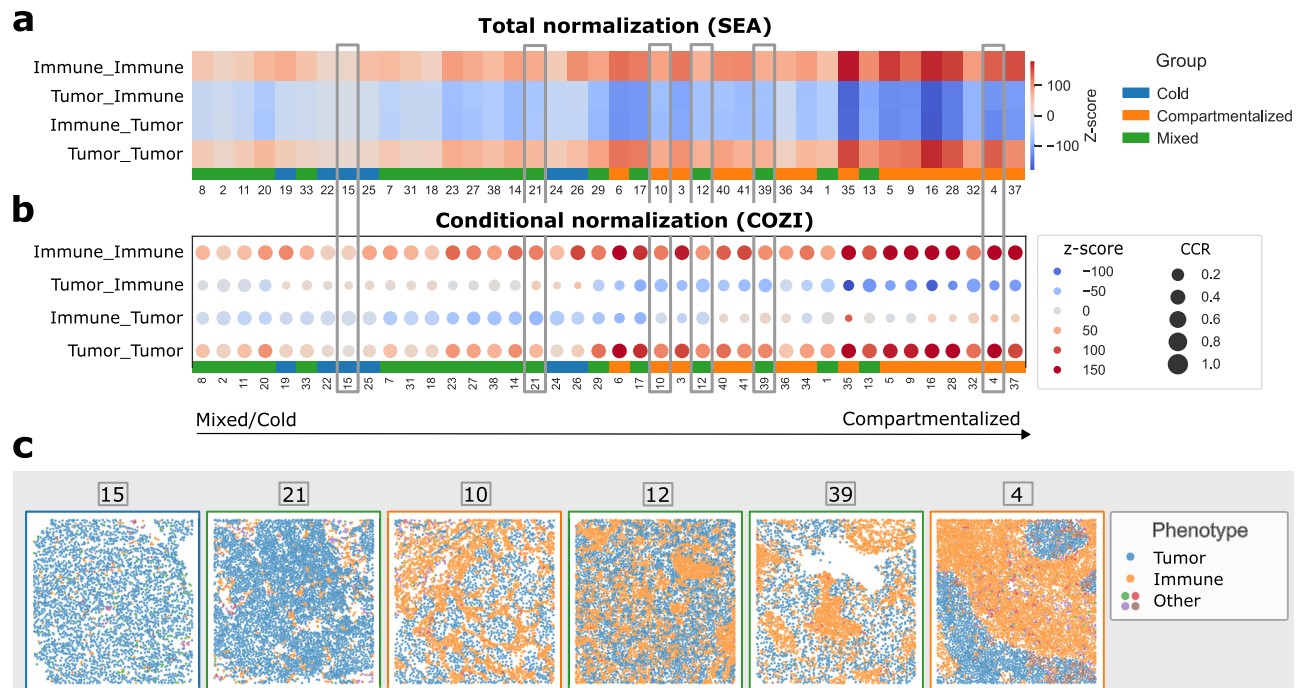

**Fig. 5 | Recovery of immune-tumor cell mixing in Triple Negative Breast Cancer.** NEP analysis of a TNBC cohort from Keren et al.[10] with $n = 40$ images. **a** SEA z-scores of interactions only involving immune and tumor cells. **b** COZI z-scores of interactions only involving immune and tumor cells. Samples are clustered based on the four COZI scores. Cohort classes (Cold, Mixed, Compartmentalized) are color coded below. **c** Exemplary images showing immune (orange) and tumor (blue) cell mixing across samples and cohorts, linked to the respective NEP results by gray lines in (**a**) and (**b**). Source data are provided as a Source Data file.

increasing purple–green preference, reflecting the shift from early infiltration (green surrounded by purple) to advanced infiltration (purple surrounded by green; Fig. 4c, I).

This directional perspective became even more relevant when comparing scenarios II and III. While total normalization yielded nearly indistinguishable results between the two datasets (Fig. 4b, II–III), COZI clearly differentiated between green cells infiltrating a purple cluster versus purple cells infiltrating a green cluster (Fig. 4c, II–III). This ability to resolve infiltration directionality is critical for interpreting biological processes where asymmetry matters, such as immune cells infiltrating tumor regions.

However, conditional normalization alone does not reveal how many cells contribute to an observed z-score. In scenario II, for instance, a very high green–purple z-score was driven by only ~10% of green cells that were strongly surrounded by purple, while the majority of green cells were not (Fig. 4d). To improve interpretability, we therefore introduce the conditional cell ratio (CCR), which quantifies the proportion of cells of a given type that contribute to the conditional interaction. By plotting CCR alongside COZI z-scores (Fig. 4e), users can distinguish whether an enrichment reflects widespread infiltration or only a small subset of cells.

In summary, total normalization provides a useful global overview of tissue architecture, whereas conditional normalization enables fine-grained and directional interpretation. Combining COZI with CCR offers both sensitivity to asymmetry and transparency regarding the proportion of contributing cells, which we consider crucial for the interpretation of complex tissue data. In the following section, we illustrate how these methodological differences impact the biological interpretation of real datasets.

### Recovery of immune-tumor cell mixing in triple negative breast cancer

To evaluate how COZI performs in real tissue, we next applied NEP analysis to the triple-negative breast cancer (TNBC) dataset from Keren

et al.[10]. In that study, patients were stratified into three cohorts based on immune–tumor mixing: Cold (low immune abundance, <250 immune cells), mixed (strong intermingling of immune and tumor cells), and compartmentalized (clear separation of immune and tumor regions). We analyzed how these mixing states are recovered using SEA (total normalization; Fig. 5a) and COZI (conditional normalization; Fig. 5b).

Both methods captured the self-preference of tumor and immune cells. SEA detected a general avoidance between immune and tumor cells, but provided no distinction between the two NEP directions. By contrast, COZI yielded directional scores that enabled more nuanced interpretation. When clustering samples based on the two immune–tumor NEP scores from COZI, we observed a gradient from cold/mixed to compartmentalized tumors (Fig. 5b). In cold and mixed tumors, immune cells strongly avoided tumor neighbors: most immune cells (~80%) clustered among themselves, while only a minority of tumor cells (~20%) preferred immune neighbors, reflecting tumor cells embedded within small immune aggregates. These findings were consistent with visual inspection (Fig. 5c, samples 15 and 21).

Toward the middle of the gradient, directional scores converged. In these cases, both immune and tumor cells showed self-preference and mutual avoidance, consistent with segregated cell populations. Interestingly, samples near the mixed/compartmentalized boundary (e.g., sample 10 labeled compartmentalized and sample 12 labeled mixed) displayed similar images and NEP profiles, illustrating how biological classifications can blur at cohort edges (Fig. 5c).

In clearly compartmentalized tumors, both SEA and COZI identified strong self-preferences of immune and tumor cells. Notably, COZI revealed subtle infiltration of a small fraction of immune cells into tumor-dense regions, whereas the reverse (tumor into immune regions) was absent. SEA only captured this trend coarsely as a global avoidance signal. Thus, COZI provided a more fine-grained, mechanistic view of infiltration events.

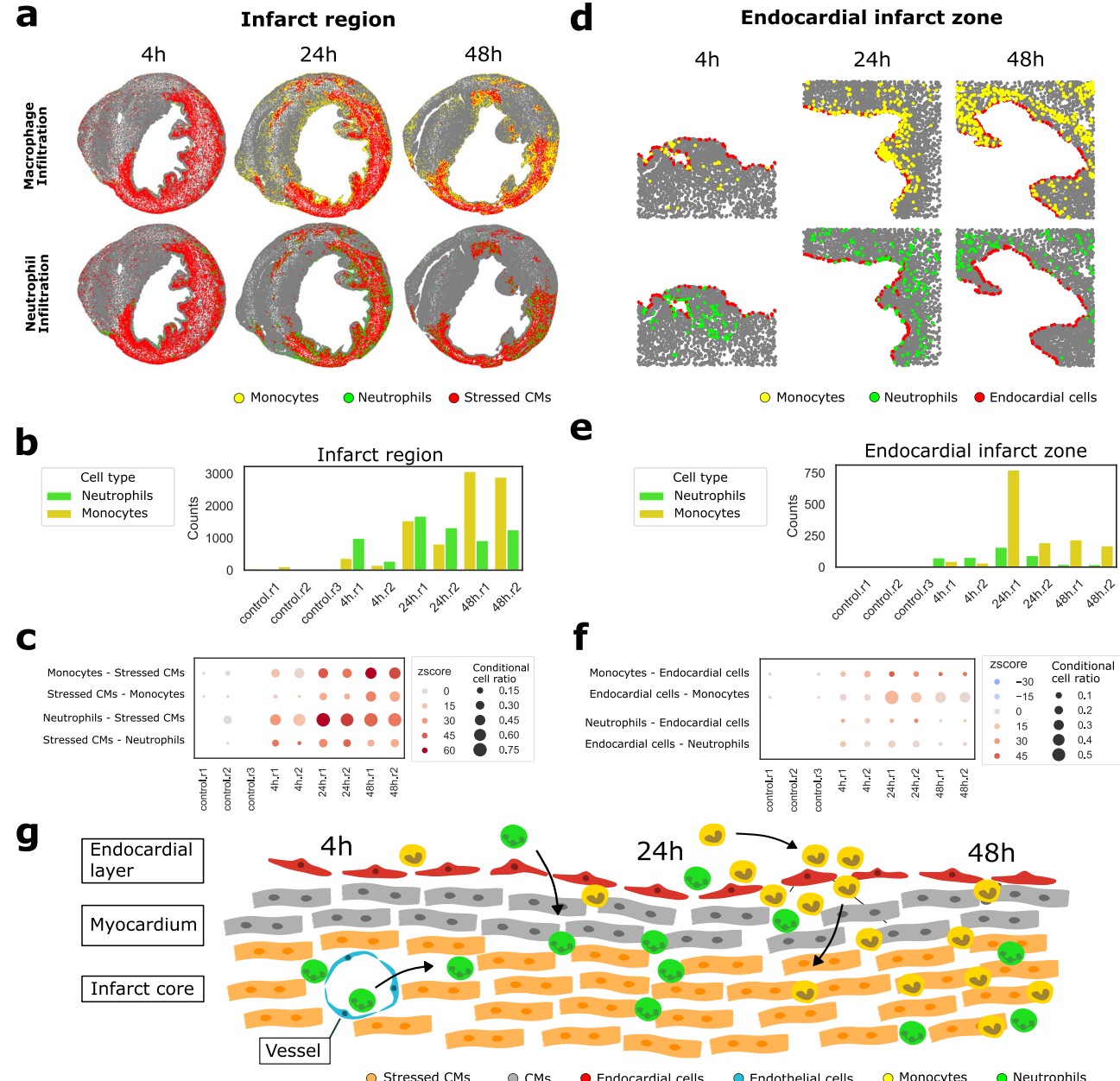

**Fig. 6 | Recovery of tissue structures in a myocardial infarction (MI) dataset.** MI mouse dataset with $n = 3$ control samples without infarct, $n = 2$ samples 4 h after, $n = 2$ samples 24 h after and $n = 2$ samples 48 h after infarct. **a, d** Monocyte (yellow) and neutrophil (green) infiltration into the (**a**) infarct region colored by stressed cardiomyocytes (CMs) (red) (4 h.r2, 24 h.r1, 48 h.r1) and through the (**d**) endocardial layer colored by endocardial cells (red) (4 h.r1, 24 h.r1, 48 h.r1) at 4, 24 and 48 h after infarct. Scatter plot colored by cell type identities. **b, e** Cell type percentages of monocytes and neutrophils in the (**b**) infarct region and the (**e**) endocardial infarct zone. **c, f** COZI neutrophil and monocyte NEP scores with (**c**) stressed CMs and (**f**) endocardial cells for control, 4, 24, and 48 h samples. **g** Schematic of biological findings for immune cell infiltration through the endocardium or blood vessels into the injury site along the timeline after infarction from left to right. Endocardial cells (red), myocardium (gray), stressed CMs (orange), endothelial cells (blue), neutrophils (green), and monocytes (yellow). Source data are provided as a Source Data file.

In summary, the TNBC dataset validates our simulation findings: SEA delivers global scores of immune–tumor organization, while COZI both recovers known cohort labels and yields interpretable directional NEP scores. Together, these results highlight COZI's utility for analyzing immune–tumor spatial relationships in cancer.

### Recovery of global and local tissue structures in a myocardial infarction dataset

We applied COZI as well as all other NEP methods to another biological dataset that did not only contain two major cell types like the TNBC dataset, but various different cell types. We analysed the myocardial infarction dataset from Wuennemann et al. 2023. The authors studied immune cell infiltration into the mouse heart infarct area as part of the healing response at multiple time points after induced myocardial infarction (4, 24, and 48 h)[25]. We used the highly multiplexed imaging (Sequential Immunofluorescence) dataset from the study, as it captures spatially resolved cell phenotypes of transverse sections of the whole mouse heart. In the data (Fig. 6a), mouse heart sections are visible with the left ventricular lumen in the center of the round cross-section. At all time points, the injured region containing Ankrd1+ stressed cardiomyocytes (stressed CMs) is visible (red, Fig. 6a). Both visual inspection (Fig. 6a) and cell count analysis of the manually

annotated infarct region by the authors (Fig. 6b) suggest that neutrophils infiltrate the injury earlier than Ccr2+ monocytes/macrophages, referred to as monocytes from here on. In addition, the original study reported that immune cell infiltration, particularly of monocytes, in the acute phase occurs through the endocardium—the inner layer of the heart—specifically in the left ventricle. These observations serve as biological anchor points to evaluate the performance of the different methods. We systematically examined the visual spatial distributions, region-specific cell type abundances, and the computed NEP scores on the whole image to guide biologically meaningful interpretations that can be applied to other datasets.

We first investigated how well the methods captured immune cell infiltration into the infarct and showed that only COZI successfully identified the earlier infiltration of neutrophils compared to monocytes while preserving directionality in NEP (Fig. 6c, Supplementary Fig. 5).

Monocyte infiltration into the infarct: Monocytes show limited infiltration into the infarct at 4 h (Fig. 6a, b). At 24 h, monocytes infiltration becomes more pronounced (Fig. 6a, b) and COZI shows a strong preference of ~40% monocytes for stressed CMs (z-score ~40, Fig. 6c), and the reverse preference is less strong and by only ~15% of stressed CMs. This observation suggests that monocytes start infiltrating the injury site by surrounding themselves with stressed CMs, while stressed CMs remain clustered together, similar to the asymmetric patterns observed in our simulations. By 48 h, monocytes are fully integrated into the infarct, leading to more similar NEP scores between the two cell types (z-score ~30–40, Fig. 6c) while also the CCR increases to ~0.4. CellCharter* and Scimap consistently capture the peak preference at 48 h as well, while they do not detect an infiltration at 24 h. CellCharter, Scimap, SEA and Squidpy all capture an avoidance in samples 4 h.r2 and 24 h.r1. At 4 h.r2, there are almost no monocytes and they visually do not seem to avoid the infarct region (Fig. 6a). At 24 h, monocytes are surrounding the infarct core visibly (Fig. 6a, Supplementary Fig. 6). Misty, histoCAT and COZI, the three methods which are capturing directionality best in the simulation experiment, detect a monocyte preference for stressed CMs at 24 h. In contrast, CellCharter, Scimap, SEA and Squidpy (Supplementary Fig. 5) report an avoidance of monocytes for stressed CMs for this time point, as they provide global scores.

Neutrophil infiltration into the infarct: Neutrophils already start infiltrating the infarct region at 4 h (Fig. 6a, c, Supplementary Fig. 6d). Looking at COZI z-scores, stressed CMs show a preference for neutrophils already at 4 h (Fig. 6c). In 4 h.r2, neutrophils form clusters of cells within the border zone of the infarct region (Fig. 6a, lower panel). Only a small fraction (<10%) of stressed CMs directly neighbor neutrophil clusters, but those interactions are unexpectedly strong, resulting in a high z-score despite a low CCR (Fig. 6c). In the opposite direction, ~50% of neutrophils are adjacent to stressed CMs, though with weaker enrichment than the stressed CM to neutrophil direction. This asymmetry produces a clear directional COZI signal. By 24 h (sample r1), neutrophils are more broadly intermixed with stressed CMs, resulting in mutual NEP scores between the two cell types (Fig. 6e, Supplementary Fig. 6d). In addition, we found that neutrophils also infiltrate through blood vessels within the infarct rather than from the surrounding tissue (Supplementary Figs. 7 and 8). This was validated by a preference of endothelial cells (which form vessel structures) for neutrophils at 4 h— a pattern again only recovered by COZI (Supplementary Figs. 7e and 8b). By 24 h, neutrophils are fully present within the infarct core, leading to mutual strong preference between the two cell types. At 48 h, neutrophils show a preference for stressed CMs, as they become fully embedded in the infarct, being surrounded by stressed CMs. When comparing the COZI scores to other NEP method results, only some capture a neutrophil infiltration already at 4 h and no other method captures a change in directionality along the time axis (Supplementary Fig. 5).

As the next step, we studied immune cell infiltration through the endocardium, a key finding of the myocardial infarction study, particularly for monocytes. We could verify the directionality of this NEP only with COZI.

Monocyte endocardium infiltration: We examined NEP scores between endocardial and immune cells over time after the infarct. The images clearly show starting infiltration at 4 h, a strong infiltration at 24 h and further infiltrated monocytes at 48 h (Fig. 6b). Looking at COZI scores at 4 h, monocytes have a slight preference for endocardial cells, while at 24 h, monocytes show a strong preference for endocardial cells (z-score ~40), and endocardial cells also exhibit an elevated preference for monocytes (z-score ~20), reflecting the peak of infiltration through the endocardium (Fig. 6d, f). At 48 h, monocytes are observed deeper within the tissue, reducing neighbor counts with endocardial cells. Endocardial cells lose their NEP for monocytes as large areas of the endocardial layer are no longer neighboring monocytes, while monocytes are migrating. All other methods capture a peak infiltration through the endocardium at 24 h, while most are largely resembling a strong increase in monocyte abundance and interaction count in sample 24 h.r1 (Supplementary Fig. 9). Additionally, CellCharter, Misty, and Scimap report a preference of endocardial cells for monocytes, in contrast to COZI, which shows the opposite direction (Supplementary Fig. 9). The lower abundance of endocardial cells compared to monocytes at 24 h highlights the method's limitation in capturing preference from highly abundant to lowly abundant cell types (Supplementary Fig. 8a). Here, combining the COZI z-score with the CCR is crucial: although monocytes show a strong apparent preference for endocardial cells, this signal is driven by fewer than 10% of monocytes, reflecting how abundance imbalances can obscure interpretation if CCR is not considered.

Neutrophil endocardium infiltration: Neutrophils show a weaker preference for endocardial cells at 4 h (z-score ~10–20) than monocytes, indicating less infiltration through the endocardium than monocytes. By 24 h, neutrophil infiltration through the endocardium increases but remains weaker than monocyte infiltration, with endocardial cells not exhibiting the same strong preference for neutrophils. This is reflected in the images and the decline in neutrophil infiltration by 48 h (Fig. 6b, d). Notably, COZI was the only method that captured neutrophil infiltration through the endocardium at a biologically meaningful level, both overall and at specific time points (Fig. 6f, and Supplementary Fig. 9).

We assessed the quality of the manually annotated regions by examining correlations between conditional cell ratios (the proportion of cells adjacent to another cell type, e.g., proportion of neutrophils adjacent to stressed CMs) and the fraction of those cells within the respective annotated regions (e.g., neutrophils in the infarct region) (Supplementary Fig. 10a–d). Strong correlations, such as between the proportion of neutrophils neighboring endocardial cells and the fraction of neutrophils in annotated endocardial infarct zones, provided independent support for the biological validity of the annotations.

To evaluate the impact of neighborhood definition on NEP results, we compared multiple neighborhood constructions across samples: Delaunay triangulation, k-nearest neighbors (k = 5, 10, 15), and fixed-radius neighborhoods (r = 50, 100, 150; Supplementary Fig. 11). Since NEP methods are designed to capture local interactions, overly large neighborhoods risk blurring immediate and distant interactions, while very small ones may capture too few neighbors. Indeed, z-scores were highly correlated (>0.9) across neighborhood definitions, with the exception of the smallest radius (r = 50), which deviated from the others (Supplementary Fig. 11a). This indicates that neighborhoods that are too small might fail to capture sufficient interaction context.

Correlations remained high when only correlating NEP score vectors of specific cell type pairs (Supplementary Fig. 11b–f), confirming the stability of COZI scores across graph definitions.

Hierarchical clustering further showed that neither neighborhood type nor size consistently dominated clustering outcomes. These results held for both broadly distributed cell types such as stressed cardiomyocytes (Supplementary Fig. 11b–d) and spatially confined types such as endocardial cells at tissue edges (Supplementary Fig. 11e–f).

We further explored myocardial immune cell infiltration plots similar as in Fig. 6c and f (k = 5), but now choosing small neighborhoods (radius 50 pixels) as well as larger neighborhoods (knn with k 15 and 30) (Supplementary Fig. 12). The results show preserved NEP scores across all three neighborhood definitions. Importantly, the conditional cell ratio as well as the range of z-scores increases with increasing neighborhood size as the number of interactions counted per index cell increases drastically. The results align with our findings from Fig. 6 showing infarct core infiltration of neutrophils before monocyte infiltration (Supplementary Fig. 12a–c). Also the monocyte NEP for Endocardial cells at 24 h is recovered across all neighborhood definitions (Supplementary Fig. 12d–f).

This indicates that the recovered NEPs are already apparent with close neighbors and either persist or do not get hidden by including more distant neighbors into the neighborhood definition. Overall, NEP findings were consistent across neighborhood definitions underscoring the robustness of both the biological finding and our approach.

## Discussion

This study provides a systematic comparison of NEP analysis methods and proposes an alternative combination of common NEP analysis steps for gaining biologically meaningful insights. This work not only highlights the heterogeneity of the available methods and their results, but also the wide range of subtle, heterogeneous spatial differences that can be recovered from spatial data. With an appropriate score, NEP methods can recover the strength of neighbor preferences, which is often the biologically relevant quantity rather than mere detectability. We also revealed that directionality is biologically meaningful yet inconsistently captured across existing approaches. This makes our study a valuable resource for researchers, helping them make informed decisions for their spatial analysis. Additionally, we demonstrate how IST-generated spatial data can effectively be used to compare spatial omics methods.

When evaluating NEP methods, we found that all methods could distinguish between tissue cohorts based on spatial differences. However, minor algorithmic differences significantly impacted the ability to recover biologically relevant spatial patterns and led to vastly heterogeneous results. All methods except histoCAT and IMCRtools classic provide normally distributed scores and can therefore better distinguish between different NEP strengths. SEA, Giotto, IMCR classic, and Squidpy provide global preference or avoidance NEP scores between cell types but do not recover NEP directionality, a key aspect of spatial organization. We demonstrated that conditional averaging, as introduced by histoCAT, is essential for recovering directionality by normalizing interaction counts exclusively among interacting cells. Our results showed that COZI, a combination of conditional normalization and a z-score, captured directional relationships in simulated and biological datasets. Although CellCharter, Misty, and Scimap also recovered directionality, these methods were sensitive to differences in cell type abundances. Specifically, they could only detect preferences from a lowly abundant cell type toward a highly abundant one, but not vice versa. Overall, throughout our systematic comparison, we identified (1) sensitivity and (2) directionality as major differentiation axes between NEP methods. Methods with improved sensitivity but no directionality recovery yield symmetric, global scores of cell-type preference or avoidance in an image. In contrast, COZI z-scores together with the CCR recover interpretable directional NEPs, enabling to study fine-grained infiltration patterns while remaining robust across varied data characteristics (e.g., imbalanced cell-type abundances and subtle spatial differences). For NEP analysis, we recommend combining symmetric global methods with COZI for directional analysis, and evaluating both results in light of the biological question being addressed. The simulated and biological examples presented here guide the interpretation of both scores.

A limitation when using COZI is that interpretation based solely on z-scores provides an incomplete view, especially in more complex tissues. Only the CCR, in combination with z-scores, enables a fine-grained interpretation of spatial images. By normalizing only over index cells that neighbor at least one cell of the partner type, conditional z-scores restrict the analysis to regions where interactions actually occur. Together with the CCR, this makes COZI an approach that bridges classical global NEP methods and local window–sliding approaches[26]. As with all spatial scores, it is important to note that NEP results are only directly comparable between cohorts when cell-type abundances are similar.

Another important discussion point are assumptions regarding tissue structure when performing whole-slide permutation of cell type labels, as used in COZI and many other NEP methods. Whole slide spatial randomization assumes that cell types are randomly distributed independently of tissue structure, an assumption that is unrealistic in many biological contexts. A good example is the presence of large tumor cell clusters, which naturally form contiguous aggregates rather than random distributions. With this thought, the recently introduced Kontextual framework by Ameen et al.[27] provides a different conceptual idea by performing context-restricted permutations. They propose to randomize "child" cell types only across their "parent" cell type areas, for example permuting CD8+ cells only within their parent immune cell compartment rather than across the entire image. In Ding et al.[28] they first identify hotspots of high diversity in tissues and only afterwards perform colocalization analysis in these hotspot patches. Both methods thereby preserve aspects of tissue architecture while still enabling statistical testing. This is a very interesting direction for the next generation of NEP methods.

Another potential future direction is the expansion of NEP analysis beyond local cell-cell neighbors, providing deeper insights into global tissue architecture. A promising approach is the combination of a conditional z-score with a multiview framework like Misty, which captures multiple spatial neighborhood views at once. While NEP methods like COZI can define larger neighborhoods, the defined space includes direct and distant neighbors at the same time. In contrast, Misty and other multiview approaches[4] allow for the independent assessment of short- and long-range spatial neighbors. Kasumi further combines the idea of short and long-range interactions of Misty with a window-sliding approach for more context dependent NEP analysis. Future work could explore how to best integrate these approaches for a more comprehensive understanding of global and local tissue organization.

Since this is the first study to perform a method comparison using the IST framework from Baker et al.[14], we also identified some limitations associated with this approach. One key limitation was simultaneously maintaining predefined cell type abundances and adjacency constraints. When a low-abundance cell type was required to maintain high adjacency with multiple cell types, there were often too few cells to comply with both conditions. During these simulations, the IST framework prioritizes adherence to the cell-cell adjacency parameter over maintaining exact cell type abundances. As a result, cell-cell adjacencies were reliably simulated while deviations from the given cell type abundances occurred. Understanding these inherent tissue architecture limitations–e.g., that a low abundant cell type cannot have very high preferences for all other cell types–can help setting realistic expectations when analyzing biological tissues. Despite these challenges, IST-generated data allowed us to identify algorithmic differences between methods that might have been overlooked in real-world datasets. One could also expand the simulation framework to 3D

simulations, addressing the increasing availability of 3D spatial omics datasets. However, the COZI score is affected by the total number of cells, which is much more variable in 3D datasets than in 2D datasets. The higher the cell number, the higher the COZI score due to the effect size of z-scores. We therefore recommend dividing COZI scores of a sample by the square root of the number of cells in that sample when encountering large differences in cell type numbers.

Finally, NEP methods are primarily tailored to single-cell resolution data, as found in spatial transcriptomics and proteomics, where neighborhoods are defined by direct spatial proximity between individual cells. However, some technologies in transcriptomics and proteomics operate at lower spatial resolution, where each spot represents a mixture of cell types. In these cases, NEP analyses could be applied at the spot-cluster level, but interpretation beyond cluster labels is limited. Although deconvolution methods can estimate cell-type proportions per spot, defining neighbor relationships and deriving reliable interaction statistics remains challenging. Conversely, at subcellular resolution, NEP concepts could be extended to individual transcript molecules, defining "transcript neighborhoods" and assessing their spatial patterns. Both spot-level and subcellular approaches introduce additional uncertainty and may require rethinking interaction scores and normalization strategies. We see this as an important future direction for methodological development and standardization, particularly as the field moves toward integrating spatial data across multiple technologies and resolutions.

Biological tissues are organized into complex spatial architectures, including cellular niches, gradients, and functional domains. Analytical approaches beyond NEP can capture spatially coherent multicellular structures and thereby provide a more comprehensive view of tissue organization, function, and disease progression. Nevertheless, statistical NEP scores such as COZI quantify a well-defined and biologically meaningful aspect of spatial organization while remaining easy to interpret and robust across datasets. In the context of ongoing efforts to translate spatial data into clinical applications, NEP scores therefore represent a practical and reliable choice. Our study contributes to this endeavor by helping to clarify which methods to use and how their results should be interpreted within specific biological contexts.

## Methods

This study used only data from previously published studies. All original data collection procedures were approved by the respective institutional review boards or ethics committees as described in the corresponding publications. Therefore, no additional ethical approval was required for this study.

### Tissue data generation

**In silico tissue generation framework.** We generated in silico tissues (IST) using the open-source IST generation method from Baker et al.[24]. We used the provided Python scripts for tissue scaffold generation and cell type label assignment (https://github.com/klarman-cell-observatory/PowerAnalysisForSpatialOmics).

First, blank tissue scaffolds were generated with a random-circle packing algorithm. Parameters for the field of view (FOV) size $f$ and $r_{max}$ and $r_{min}$ radius of cells were varied. In the final dataset, we used $f = 1000$, $r_{min} = 10$, and $r_{max} = 10$. The circle representation was converted into a Voronoi graph representation for downstream cell type annotation. The script was adapted to output the tissue matrices of the scaffolds for further annotation. The nodes of the unlabeled tissue scaffold were annotated using the heuristic approach due to speed and practicability. A vector $p$ containing the abundances for each of the $k$ cell types was specified.

A $k \times k$ matrix $H_{k \times k}$ defines the probability that a cell $k_i$ is adjacent to a cell $k_j$. In the original method's use case, $p$ and $H_{k \times k}$ were estimated from the provided biological tissue. Here, we provided the parameters to generate tissue architecture cohorts with similar cell type abundances. The image was partitioned into a grid of regions with size t x t pixels for heuristic annotation. We used a grid size of 200. Within an initial random region of blank nodes, the cell type labels were sampled from a multinomial distribution, with $p_k$ being the cell type distribution. The neighborhood graph distance was set to 2. Depending on $k$, the neighboring node identities were sampled from a multinomial distribution of the respective row vector of the adjacency matrix $H_{ki \times k}$. The region of the grid partition was shifted by $t/2$ in the x and y direction, and the annotation was repeated for every unlabeled node. After completion, random nodes were selected, and the neighborhood composition was calculated and compared with $H_{k \times k}$. Nodes with overabundant cell type labels were swapped to under-abundant ones. The number of final iterations $I$ was 300 and cell type label swaps $S$ was 50.

We simulated cohorts with differing cell-cell adjacencies and cell type abundances of 4 cell types (Table 1). The cell types were red (0), yellow (1), green (2) and purple (3). We will use the numbers 0-3 in the method section while we used the respective colors in the result section for comparability to the images. Across cohorts, we used the same tissue scaffolds for cell type annotations. We adapted $H_{k \times k}$ and $p$ to generate one symmetric dataset D1 (Table 2), and an asymmetric dataset, D2 (Table 3). Per dataset, we generated three tissue architecture levels for cohort generation. We simulated D1 with a random ($H_{0x0} = 0.25$), a weak ($H_{0x0} = 0.45$), and a strong ($H_{0x0} = 0.6$) self-preference of cell types 0 (Table 2). For D2, we simulated a random ($H_{0x1} = 0.25$), a weak ($H_{0x1} = 0.45$), and a strong ($H_{0x1} = 0.6$) cross-preference between cell type 0 and 1 (Table 3). Per adjacency level, the abundance of cell type 0 $p_{k=0}$ ranged from 0.05 to 0.55, with $p_{k=\{1, 2, 3\}} = (1-p_{k=0})/3$ (Table 1). Per dataset (D1 and D2), we combined the eight vectors for $p$ with the three different adjacency matrices $H_{k \times k}$ for cell type label assignment. Therefore, we generated 24 different cohorts with 100 samples each per dataset.

Of note, IST generation has local tissue organization constraints and cannot build global relationships in a sample. However, the NEP methods we compared are restricted to local tissue organization and thus are well suited to this setup. We further performed quality control of the simulated data and identified discrepancies between the simulated and the observed cell type abundance levels (Supplementary Fig. 1a and b, Methods). This discrepancy was most apparent in cohorts with lowly abundant red cell type or very highly abundant red cell type but not apparent in cohorts with similar cell type abundances across cell types. Therefore, our study revealed critical limitations in the published framework.

**SpaSim tissue simulations.** Cell infiltration tissue simulations were generated with SpaSim. All simulations were based on a common

## Table 1 | Cell type abundance vectors $p_{k1-4}$ for ranging cell type abundances of cell type 0 for D1 and D2

| Cell type | ab0_0.05 | ab0_0.1 | ab0_0.15 | ab0_0.2 | ab0_0.25 | ab0_0.35 | ab0_0.45 | ab0_0.55 |
|---|---|---|---|---|---|---|---|---|
| $p_0$ | 0.05 | 0.1 | 0.15 | 0.2 | 0.25 | 0.35 | 0.45 | 0.55 |
| $p_1$ | 0.32 | 0.3 | 0.28 | 0.27 | 0.25 | 0.22 | 0.18 | 0.15 |
| $p_2$ | 0.32 | 0.3 | 0.28 | 0.27 | 0.25 | 0.22 | 0.18 | 0.15 |
| $p_3$ | 0.31 | 0.3 | 0.29 | 0.26 | 0.25 | 0.21 | 0.19 | 0.15 |

**Table 2 | Adjacency matrices H$_{kxk}$ for investigating self-preference of cell type 0 in D1 on three different levels**

| Random | | | | |
|---|---|---|---|---|
| | $k_0$ | $k_1$ | $k_2$ | $k_3$ |
| $k_0$ | 0.25 | 0.25 | 0.25 | 0.25 |
| $k_1$ | 0.25 | 0.25 | 0.25 | 0.25 |
| $k_2$ | 0.25 | 0.25 | 0.25 | 0.25 |
| $k_3$ | 0.25 | 0.25 | 0.25 | 0.25 |
| Weak self-preference 0-0 | | | | |
| $k_0$ | 0.45 | 0.18 | 0.18 | 0.19 |
| $k_1$ | 0.18 | 0.27 | 0.28 | 0.27 |
| $k_2$ | 0.18 | 0.28 | 0.27 | 0.27 |
| $k_3$ | 0.19 | 0.27 | 0.27 | 0.27 |
| Strong self-preference 0-0 | | | | |
| $k_0$ | 0.60 | 0.13 | 0.13 | 0.14 |
| $k_1$ | 0.13 | 0.29 | 0.29 | 0.29 |
| $k_2$ | 0.13 | 0.29 | 0.29 | 0.29 |
| $k_3$ | 0.14 | 0.29 | 0.29 | 0.28 |

**Table 3 | Adjacency matrices H$_{kxk}$ for investigating cross-preference of cell type 0 to cell type 1 in D2 on three different levels**

| Random | | | | |
|---|---|---|---|---|
| | $k_0$ | $k_1$ | $k_2$ | $k_3$ |
| $k_0$ | 0.25 | 0.25 | 0.25 | 0.25 |
| $k_1$ | 0.25 | 0.25 | 0.25 | 0.25 |
| $k_2$ | 0.25 | 0.25 | 0.25 | 0.25 |
| $k_3$ | 0.25 | 0.25 | 0.25 | 0.25 |
| Weak cross-preference 0-1 | | | | |
| $k_0$ | 0.18 | 0.45 | 0.18 | 0.19 |
| $k_1$ | 0.27 | 0.18 | 0.28 | 0.27 |
| $k_2$ | 0.28 | 0.18 | 0.27 | 0.27 |
| $k_3$ | 0.27 | 0.19 | 0.27 | 0.27 |
| Strong cross-preference 0-1 | | | | |
| $k_0$ | 0.13 | 0.60 | 0.13 | 0.14 |
| $k_1$ | 0.29 | 0.13 | 0.29 | 0.29 |
| $k_2$ | 0.29 | 0.13 | 0.29 | 0.29 |
| $k_3$ | 0.29 | 0.14 | 0.29 | 0.28 |

background consisting of 5000 "other" cells (width = 2000, height = 2000, method = "Hardcore", min_d = 10, oversampling_rate = 1.6). This background served as the basis to which cell clusters were added by relabeling subsets of cells.

Dataset I contained a single oval-shaped purple cluster of 500 cells. Green cells were progressively infiltrated into the purple cluster at proportions of 0.1, 0.2, 0.3, 0.4, 0.5, 0.6, and 0.7 (pI).

Dataset II contained two clusters, one purple (500 cells) and one green. To model infiltration, increasing proportions of green cells were placed into the purple cluster (pII = 0, pI). To maintain equal overall abundances of purple and green cells, the size of the green cluster was gradually reduced across simulations (510, 500, 490, 480, 470, 460, 450, 440 cells).

Dataset III was generated analogously to Dataset II but with the purple and green labels switched.

### Neighbor preference (NEP) methods

Most NEP methods compute statistical scores to quantify neighbor preferences or avoidance. These include permutation-based

significance testing (e.g., in histoCAT and IMCRtools classic) or standardized z-scores (e.g., Scimap, SEA, COZI). We used each method's default statistical test and interpretation logic as provided by the authors and did not include intermediate output scores. We did not filter interactions based on the calculated p-values, but instead analyzed the interpretable scores output by each method. The NEP methods used do not assume normality of the data. In simulated and real datasets, tissue images or regions were treated as independent samples.

**histoCAT and IMCRtools.** The IMCRtools R toolbox (v1.8.0) provides the countInteractions() and testInteractions() functions for NEP analysis with method options "classic" and "histoCAT" to find statistically enriched NEPs of cell types in defined cellular neighborhoods (https://github.com/BodenmillerGroup/imcRtools). The original neighborhood definition of histoCAT is distance-based[8]. We performed the classic and histoCAT testInteractions() analysis of simulated data with a Delaunay graph neighborhood definition, as provided in IMCRtools for comparability. For the biological dataset we used k = 5. NEPs are counted bi-directionally, so the method provides preference values for cell type A to type B and from type B to type A. The neighbor count is averaged in two ways: the classic method divides the interaction count between A and B by the total number of cells of type A in the sample, while HistoCAT divides the interactions between A and B by the number of type A cells with at least one neighbor of type B[8]. We termed this "conditional averaging" throughout the manuscript. After counting, the cell type labels are randomly permuted ($n = 300$ times), and randomized interactions are counted and averaged as described. If an interaction is abundant significantly more or less is indicated in the NEP score "sigval" with 1 and −1 respectively. Not significant interactions are reported as 0. We obtained a sample-by-interaction matrix with sigval values as output for downstream evaluation.

**Giotto.** The Giotto NEP analysis was performed with the cellProximityEnrichment() function in the Giotto R package. The method defines neighbors by Delaunay triangulation. Giotto counts interactions non-directionally, providing only one value for the interaction A-B. We permuted cell type labels randomly $n = 300$ times, forming a null distribution. The true interaction count is divided by the mean null distribution count and termed cell proximity score (CPscore). A wrapper of the function is written in cellProximityEnrichment() of the package. We obtained a sample-by-interaction matrix with CPscores as output for downstream evaluation.

**Spatial enrichment analysis (SEA).** Spatial enrichment analysis (SEA) detects statistically enriched cell-cell interactions[10]. The method is described and used in several papers[10,14] but not provided as a standalone package. The authors kindly shared Matlab and Python code for analysis, which we adapted on a fork of Scimap to use their underlying neighborhood definition and permutation framework. The method was initially described using an Euclidean distance neighborhood definition. We implemented the method with a bi-directional count, while non- and bi-directional counts were both described in studies performing SEA. We ran the method with a Delaunay graph neighborhood definition for the simulated and knn (k = 5) for the biological dataset and permuted the cell type labels $n = 300$ times. The z-score is calculated as:

$$Z = \frac{X - \mu}{\sigma} \qquad (1)$$

Where:

$X$ = observed neighbor count
$\mu$ = mean of null distribution of X in permutation
$\sigma$ = standard deviation of null distribution of X in permutation

We obtained a sample-by-interaction matrix with z-scores as output for downstream evaluation.

**MISTy.** The Multiview Intercellular SpaTial modeling framework (MISTy)[9] enables analysis of the relationships of markers or cell types in spatially resolved data. The machine learning framework allows flexible definitions of spatial contexts (views). These views can, for example, be the inside of a cell (intraview), the direct surrounding of a cell (juxtaview), or a larger environment around a cell (paraview). Misty was not primarily designed for cell type level NEP analysis but can be adapted for this purpose by bypassing intraview calculations. MISTy models the complete tissue interactions for predictor/target relationships in each view. Each view-specific model quantifies the variance explained by each marker within that view for predicting the presence of the target cell type or marker. MISTy is implemented in R (Version 1.6.1.), and we adapted the script for Fig. 2 in the original publication for our analysis[9,29] (https://github.com/saezlab/misty_pipelines). We used MISTy with cell type annotations only (one-hot encoded cell types), which bypasses the intraview modeling. We defined the immediate neighborhood as juxtaview with a neighborhood threshold of 40 for Delaunay triangulation for the simulated and with knn (k = 5) for the biological dataset. We ran MISTy on the defined juxtaview with the run_misty() function, setting the argument bypass.intra to TRUE. We did not summarize the results but performed a sample-wise analysis. We obtained a sample-by-predictor/target matrix with model variances explained as NEP scores as output for downstream evaluation.

**Squidpy.** The spatial analysis suite Squidpy provides a nhood_enrichment() function for calculating NEPs for cell types in tissues. It follows the same approach as SEA and randomly permutes cell type labels to generate a null distribution for calculating z-scores. It divides the counted interactions by the total number of cells and generates a symmetric NEP score. Squidpy outputs the counted interactions, which we used for our interaction count comparison, as well as the calculated z-scores, which we used as NEP from Squidpy. We ran the method with a Delaunay graph neighborhood definition for the simulated and knn (k = 5) for the biological dataset and permuted the cell type labels $n = 300$ times. We obtained a sample-by-z-score matrix as output for downstream evaluation of Squidpy and a sample-by-count matrix as output for downstream evaluation of the interaction counts.

**Scimap.** The spatial analysis suite Scimap provides a spatial_interaction() function for calculating NEPs of cell types in tissues. The method also performs permutation tests to infer significant neighbor preferences in the tissue. While the method calculates p-values based on absolute difference or z-score-based significance, the output score is not the calculated z-score. The Scimap interaction score is the cell type abundance normalized and scaled number of interactions in the tissue. Therefore, the score does not include information from the permutation test, but significant interactions can be filtered based on the provided p-values. The original function only allows for the definition of k-nearest neighbors or Euclidean distance-based neighborhoods. We implemented a Delaunay graph neighborhood definition to compare it to the other methods. We ran the method with a Delaunay graph neighborhood definition for the simulated and knn (k = 5) for the biological dataset and permuted the cell type labels $n = 300$ times. We obtained a sample-by-score matrix as output for downstream evaluation.

**CellCharter.** CellCharter is a spatial analysis suite and offers an nhood_enrichment() function for computing NEPs between clusters in a spatial graph. In the publication, this function is proposed for NEPs between cells of niches, while we ran the function for NEPs between

cells of phenotypes without changing anything in the function. The method does not perform permutation testing but, in the asymmetric version that we used, calculates the difference between the observed and the expected neighbor edges in an analytical formula. We ran the method with only_inter=False as CellCharter and only_inter=True as CellCharter* to either include or exclude links between homotypic clusters, respectively[11]. We ran the method with Delaunay 1-hop neighborhood definition for the simulated and the biological dataset. We obtained a sample-by-score matrix as output for downstream evaluation.

**Conditional z-score (COZI).** The conditional z-score (COZI) outputs normalized z-scores as NEP scores. Let $C_i$ and $C_j$ be subsets of $C$, where each $C_k$ represents all cells of a specific cell type $k$ in the image. The boolean adjacency matrix $A$ contains all neighboring cell edges, where $A_{st}$ indicates the presence of an edge between cells s and t.

The observed number of edges between $C_i$ and $C_j$ is given by $N$ (Formula 2).

$$N = \sum_{s \in C_i, t \in C_j} A_{st} \tag{2}$$

To account for cell abundances, we normalize the observed interactions $N$ by $M$, the number of cells of type $C_i$ with at least one edge to a cell of type $C_j$ (Formula 3)

$$M = \sum_{s \in C_i} \mathrm{sgn}\left( \sum_{t \in C_j} A_{st} \right) \tag{3}$$

This results in the normalized interaction count $O$ (Formula 4).

$$O = \frac{N}{M} \tag{4}$$

To assess the significance of observed interactions, COZI performs permutation testing by randomly shuffling the cell type labels across cell locations. In each permutation, neighbor counts are recomputed using the same normalization procedure to obtain a null distribution for $O$.

The final COZI score is the z-score, of the observed $O$ relative to its permuted distribution, using the mean $\mu_{perm}$ and standard deviation $\sigma_{perm}$ of $O_{perm}$ across permutations (Formula 5).

$$COZI = \frac{O - \mu_{perm}}{\sigma_{perm}} \tag{5}$$

A high COZI score indicates stronger-than-random spatial association, while a low or negative score suggests no significant spatial preference or avoidance. COZI is affected by differences in total cell type numbers. The scores are higher with higher cell counts. If there are vast differences in cell type numbers in a dataset to analyze, which was not the case in our study, we recommend dividing COZI scores of a sample by the square root of the number of cells in that sample.

In addition to COZI z-scores, we also introduce the conditional cell ratio (CCR) (Formula 6).

$$CCR\left(C_i \mid C_j\right) = \frac{M}{|C_i|} \tag{6}$$

This ratio describes the proportion of cells of type $C_i$ that have at least one neighboring cell of type $C_j$.

We implemented COZI on a fork of Scimap to leverage the existing neighborhood and permutation test framework developed by labsyspharm. Notably, the method could have been implemented using other spatial analysis suites (e.g., Squidpy, IMCRtools) as well, since most perform the same underlying analysis steps. We adapted the spatial_interaction() function of Scimap to normalize the number of interactions "conditional" and also output the calculated z-score.

Therefore, the method follows the SEA approach of calculating a z-score with another interaction count averaging. We ran the method with a Delaunay graph neighborhood definition for the simulated and knn (k = 5) for the biological dataset and permuted the cell type labels $n = 300$ times. We obtained a sample-by-z-score matrix as output for downstream evaluation.

## Method comparison

### Systematic comparison with simulated data

**Cohort-level comparison with in silico tissue simulated data.** Our basic rationale for systematic method comparison was to compare the abilities to distinguish between different cohorts based on tissue architecture. Therefore, we compared between different cell-cell adjacency groups (random, weak, strong) with similar cell type abundances.

We quantitatively compared the NEP methods on simulated data by determining how well cohorts were classified and how tissue architecture was recovered. The measured features were cell-cell adjacencies (e.g., 0-0, 0-1,...). We trained a random forest model (*caret* package in R) per cohort comparison with the feature vectors per sample. We did not aim to find the ideal model to perform cohort distinction but to identify classification performance differences between the methods. We performed an 80/20 train-validation split with 5-fold cross-validation.

We applied the F1-score as a quantitative comparison measure. The F1-score is the harmonic mean between precision and recall (Formula 7).

$$F1Score = 2 \times \frac{TP}{2 \times TP + FP + FN} \quad (7)$$

Where:

$TP$ = true positive values of classification
$FP$ = false positive values of classification
$FN$ = false negative values of classification

An F1-score of 1 is the best possible result, while in a binary classification, an F1-score of 0.5 is random. We also generated a baseline for classifying the cohorts based on cell type abundances. And we added a random model where we randomly shuffled the cohort labels.

We evaluated whether the true tissue features were used for cohort distinction by assessing the cosine similarity between the ground truth and the result vectors (Formula 8).

$$\text{Cosine Similarity} = \cos(\theta) = \frac{\sum_{i=1}^{n} A_i B_i}{\sqrt{\sum_{i=1}^{n} A_i^2} \sqrt{\sum_{i=1}^{n} B_i^2}} \quad (8)$$

Where:

$A_i$ = $i$-th components of ground truth vector $A$ of NEPs
$B_i$ = $i$-th components of NEP result vector $B$ of NEPs

The ground-truth vector $A$ was created by vectorizing the cell-cell adjacency matrices (Tables 2 and 3). We subtracted the two ground-truth cell-cell adjacency vectors that were compared to generate the ground-truth comparison vector. The result vector $B$ included the extracted feature importances of the comparison-specific trained random forest model. The cosine similarity was scaled to a scale between 0 and 1, 1 being perfect similarity and 0 being dissimilarity (Formula 9).

$$\text{Scaled Cosine Similarity} = \frac{\cos(\theta) + 1}{2} \quad (9)$$

For Giotto, the only tool providing one score per neighbor preference pair, we copied the given scores for an NEP A-B to represent B-A. Per comparison, we determined 100 F1 and 100 cosine similarity scores and plotted them per cell-cell adjacency and cell-type abundance group per tool. Both scores can be compared across methods.

**Sample-level comparison with SpaSim simulated data.** We analyzed Datasets I-III with COZI and SEA. We plotted the z-scores as heatmaps and in a second step the z-scores together with the CCR as dotplots for COZI.

### Performance comparison on biological data

**Triple negative breast cancer (TNBC) dataset.** We used the TNBC dataset from Keren et al.[10]. The 40 samples were analyzed with COZI and SEA with a k = 5 k-nearest neighbor neighborhood definition. We depicted z-scores for SEA as heatmap and z-scores together with CCR for COZI as dotplot. We hierarchically clustered the samples based on COZI z-scores of the Immune_Keratin-positive tumor and Keratin-positive tumor_Immune interactions. We colored the samples based on the author annotations of Cold, Mixed and Compartmentalized tumors.

**Myocardial infarction dataset.** We ran all described NEP methods on the mouse myocardial infarction dataset by Wuennemann et al.[25] studying immune cell infiltration into the infarct core through the endocardial layer. We used the provided cell type labels for NEP analysis, excluding cell types with label "exclude". We used a k = 5 k-nearest neighbor neighborhood definition. Otherwise, we used the Delaunay neighborhood definition.

We showed cell type abundances in specific regions in the manuscript. We selected the infarct core and the two border zones for the region "infarct region" and the region Endocardial region for the region "Endocardial infarct zone". We correlated the CCR of neutrophils or monocytes with Endocardial cells with the percentage of the respective cells in the endocardial infarct zone and the CCR of neutrophils or monocytes with stressed CMs with the percentage of the respective cells in the infarct region with pearson correlation. We evaluated the effect of neighborhood size choice by correlating the COZI NEP score outputs of different cell type pairs across various neighborhood definitions (Supplementary Fig. 11). Further, we report results of small and large neighborhood sizes used in the Myocardial infarction dataset to explore the robustness of the findings described in the manuscript (Supplementary Fig. 12).

For visualization of neutrophil infiltration (Supplementary Fig. 8c), we used a crop of a representative autofluorescence-subtracted multiplexed immunofluorescence image from Wünnemann et al.[25] Full details of image acquisition and processing are reported in the original publication.[25] The image was loaded from an OME-TIFF file in Python using tifffile (v2025.2.12), zarr (v2.13.3), and dask (v2024.6.2). DAPI (channel index 0), CD31 (channel index 5), and MPO (channel index 14) were selected and rendered in cyan, yellow, and magenta respectively. Each channel was linearly normalized to the 0-1 range using the full 16-bit acquisition range (0-65535), and no gamma correction was applied. The composite RGB image was assembled and exported at 300 DPI using matplotlib (v3.9.0) and scikit-image (v0.22). Arrows annotating structures of interest were added post-acquisition using Inkscape.

**Statistical Visualization.** Box plots show the median (center line), the 25th and 75th percentiles (box limits), and the whiskers extend to 1.5 times the interquartile range (IQR) or the most extreme data point within that range. Outliers, which are points beyond the whiskers, are shown as individual points.

## Reporting summary

Further information on research design is available in the Nature Portfolio Reporting Summary linked to this article.

## Data availability

The IST data simulated in this study is deposited in (https://github.com/SchapiroLabor/NEP_comparison/simulated_data) and can be reproduced with the provided scripts in (https://github.com/SchapiroLabor/IST_generation_SCNA). The SpaSim data simulated in this study is deposited in (https://github.com/SchapiroLabor/NEP_comparison/simulated_data) and can be reproduced with the provided scripts in (https://github.com/SchapiroLabor/NEP_SpaSim). The MI Sequential Immunofluorescence data used in this study is available via Synapse (project SynID: syn51449054): (https://www.synapse.org/Synapse:syn51449054). The dataframe with phenotypes is available via Synapse: (https://www.synapse.org/Synapse:syn65487454). The TNBC data used in this study can be found at (https://www.angelolab.com/mibi-data). We used the processed data from (https://github.com/psl-schaefer/report/tree/master/data). Source data are provided with this paper.

## Code availability

All code used to create the results in this study are available on GitHub with the specific code versions connected to this manuscript archived on Zenodo. In the following, we will provide the github repository links with the Zenodo version as citations in the references. The GitHub repository for systematic tool comparison is (https://github.com/SchapiroLabor/NEP_comparison[30]). COZI is available for use in the python package COZIpy (https://pypi.org/project/cozipy/)[31], the R package coziR (https://github.com/SchapiroLabor/coziR)[32] as well as in IMCRtools from version 1.15.3 onwards when choosing the "conditional" method. The github repositories for simulated data generation are (https://github.com/SchapiroLabor/NEP_IST_generation) (IST data)[33], and (https://github.com/SchapiroLabor/NEP_SpaSim) (SpaSim)[34]. The separate GitHub repositories for running all NEP methods are (https://github.com/SchapiroLabor/NEP_Giotto) (Giotto)[35], (https://github.com/SchapiroLabor/NEP_IMCRtools) (IMCRtools classic and HistoCAT)[36], (https://github.com/SchapiroLabor/NEP_MistyR) (Misty)[37], (https://github.com/SchapiroLabor/NEP_Squidpy) (Squidpy, CellCharter, neighbor counts and cell type abundances)[38], and (https://github.com/SchapiroLabor/NEP_scimap) (COZI, SEA and Scimap)[39]. The fork of Scimap where COZI is implemented and was used for the results of this study is (https://github.com/SchapiroLabor/scimap_COZI) (v0.2.0). All repositories contain scripts to reproduce the presented results in this study. IST computations were performed using the Baden-Wurttemberg High Performance Cluster bwForCluster Helix.

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

## Acknowledgements

We thank Ethan Baker for his input on the in silico tissue generation. We are very grateful to Leeat Keren for sharing her code for SEA. We thank Philipp Schäfer for providing the processed TNBC data. We thank Leonie Küchenhoff, Gesa Voigt and Lukas Hatscher for their input throughout the study and Victor Perez for revising the final manuscript. We acknowledge the use of OpenAI's ChatGPT in this project which assisted by adapting the language to improve clarity and coherence while ensuring that the original scientific content remained intact.

C.S. is supported by the Bruno and Helene Jöster Stiftung. C.S., M.I. and K.B. are supported by the German Federal Ministry of Education and Research (BMBF 01ZZ2004). J.T. is supported by the Ministry for Science, Research and Science Baden-Württemberg "MULTI-SPACE", the Bruno and Helene Jöster Stiftung and the Multi-dimensionAI project (CZS-Project number: P2022-08-101) made possible by funding from the Carl-Zeiss-Stiftung. D.S. is supported by the German Federal Ministry of Education and Research (BMBF 01ZZ2004); the Ministry for Science, Research and Science Baden-Württemberg "AI Health Innovation Cluster" and "MULTI-SPACE"; research funding from Cellzome, a GSK company, the Bruno and Helene Jöster Stiftung, IFOM ETS Institutional Funds and the Multi-dimensionAI project (CZS-Project number: P2022-08-101) was made possible by funding from the Carl-Zeiss-Stiftung. The authors gratefully acknowledge the data storage service SDS@hd supported by the Ministry of Science, Research and the Arts Baden-Württemberg (MWK), support by the state of Baden-Württemberg through bwHPC and the German Research Foundation (DFG) through grant INST 35/1803-1 FUGG and INST 35/1804-1 LAGG and INST 1028 35/1597-1 FUGG. For the publication fee we acknowledge financial support by Heidelberg University.

## Author contributions

C.S., M.I. and D.S. conceived and designed the study. C.S. simulated the data, implemented the methods, designed and performed the method comparison. M.I and C.S. performed Misty analysis. K.B. and C.S. created plots and analysis, and K.B. provided input for the biological MI dataset. M.I., J.T. and D.S. supervised the study. C.S. and D.S. wrote the manuscript, revised by M.I., K.B. and J.T. All authors read, discussed and approved the final manuscript.

## Funding

## Competing interests

D.S. reports funding from GSK and received fees/honoraria from Alpenglow, ariadne.ai, GSK, Immunai, Lunaphore, and Noetik. K.B. reports fees from Lunaphore. All other authors do not report any competing interests.
