## [Transparent Peer Review file · Nature Communications]

Comparison and Optimization of Cellular Neighbor Preference Methods for Quantitative Tissue Analysis

Corresponding Author: Professor Denis Schapiro

Version 0:

Reviewer comments:

Reviewer #1

(Remarks to the Author)

The authors have made significant effort to revise and, as a result, improves the quality of the manuscript on several technical aspects, such as GitHub. However, several important weaknesses persist, that is: (1). the need for a benchmark analysis of NEP analysis methods is unclear. As stated by the authors, "For an index cell (e.g., an immune or tumor cell), NEP characterizes the composition of direct neighbors and assesses whether the observed pattern of neighbors deviates from what would be expected under a spatially randomized null model.", but distinguishing random from non-random patterns is not a difficult task, as shown in Fig. 1. (2). The new method proposed here, COZI, does not appear to outperform existing methods.

(Remarks on code availability)

Reviewer #2

(Remarks to the Author)

The authors have responded effectively to my major points. The addition to the discussion addressing the challenge of spot-based technologies is welcome. It should be noted that not all these technologies could be described as 'lower' resolution (e.g. Stereo-Seq or Visium HD have individual bins that could provide subcellular resolution), but an additional step would be required to approach data similar to the single-cell data used previously (e.g. the use of software such as bin2cell to estimate cell boundaries).

Additionally, I appreciate the greater clarity the authors have provided for the mouse data used in the study. The additional description provided makes it easier for the reader to understand the exact data used.

Finally, the comprehensive response to the other reviewers' points have improved the manuscript – particularly the additional analysis provided in figure 4 using SpaSim data simulations, which provides welcome additional evidence of the effectiveness of COZI.

I'm not sure the clarification added to the manuscript with regards to single-sample vs multi-sample comparisons is very helpful to the uninitiated reader (lines 201-203). While I understand the authors' perspective regarding applications in basic vs translational biology, I think this addition could certainly be strengthened.

(Remarks on code availability)

I have no additional comments on the code at this stage.

Reviewer #3

(Remarks to the Author)

For the most part, the authors have addressed reviewers' comments well and thoroughly. Below are followups to my comments:

1. It's fair to say that unrealistic assumptions are required to derive closed form expressions of the mean and variance of COZI. However, is permutation really a less unrealistic assumption? Randomizing the cell type labels also assumes that labels are randomly distributed independently of tissue structure as the null hypothesis.

2. While correlations shown in Supplementary Figure 11 are generally high, some of them are not very high. Some of the spatial neighborhood graphs may have different biological meanings. For example, triangulation, knn with small k, and distance-based neighbors with a small radius give closer neighborhoods, while knn with larger k and distance-based neighbors with a larger radius include more distant neighbors. When the different graphs give different results, would the different results give interesting different aspects of biology? It might be interesting to illustrate this on the myocardial infarction or the breast cancer example, with a graph with only close neighbors and another with some more distant neighbors.

3. In the caption of Supplementary Figure 6: "4h and 24h after infiltration" -- did you mean "infarction"?

(Remarks on code availability)

I still think having a stand-alone package for COZI is better than the pull requests, so it's more modular. Users who want to use COZI but not other functionalities of the heavier packages where you submitted pull requests don't have to install the functionalities they don't need. For example, more people analyzing spatial transcriptomics data use Seurat rather than imcRtools.

Version 1:

Reviewer comments:

Reviewer #1

(Remarks to the Author)

This revision does not present any new evidence to further support the significance of NEP benchmark analysis or to demonstrate the additional power of COZI compared to existing methods. Therefore, my original criticisms still stand.

(Remarks on code availability)

Reviewer #3

(Remarks to the Author)

The authors have satisfactorily addressed remaining comments regarding the manuscript.

(Remarks on code availability)

While having the pip package is nice, I don't think having COZI as part of imcRtools is a good idea. While COZI is of great interest to spatial transcriptomics, imcRtools is rarely used in transcriptomics. Having a stand alone R package can increase the visibility of COZI.

REVIEWER COMMENTS

Reviewer #1 (Remarks to the Author):

The authors have made significant effort to revise and, as a result, improves the quality of the manuscript on several technical aspects, such as GitHub. However, several important weaknesses persist, that is:

(1). the need for a benchmark analysis of NEP analysis methods is unclear. As stated by the authors, "For an index cell (e.g., an immune or tumor cell), NEP characterizes the composition of direct neighbors and assesses whether the observed pattern of neighbors deviates from what would be expected under a spatially randomized null model.", but distinguishing random from non-random patterns is not a difficult task, as shown in Fig. 1.

ANSWER: We thank the reviewer for raising this point, which we are happy to further clarify for the readers. We agree that, in a highly idealized setting, separating a strongly structured from a fully randomized pattern can be straightforward as shown in our first simulation experiment in Fig. 2. However, in practice, NEP analysis is rarely performed on such extreme cases: tissues typically show subtle and heterogeneous spatial differences which, once analysed with different NEP analysis methods, provide markedly different results for the same biological question. This effect is being explored by our manuscript from Figure 3 onwards. The lack of systematic comparison, together with the heterogeneous outputs we observed across methods in multiple datasets, motivated us to perform this comparison.

Our comparison clarifies that NEP analysis is not limited to a binary "random vs. non-random" decision:

- 1) With an appropriate score, NEP methods can recover the strength of neighbor preferences (e.g., weak vs. strong interactions; Fig. 2c), which is often the biologically relevant quantity rather than mere detectability. Methodologically, they recover these scores by comparing the observed to randomly permuted cell type labels.
- 2) Our systematic comparison revealed that directionality (asymmetric neighbor preferences) is biologically meaningful yet inconsistently captured across existing approaches.

Taken together, while the conceptual task may appear simple, our results show that NEP inference is not trivial in practice, because method-specific choices lead to divergent quantitative outputs and biological interpretations. This is particularly relevant in spatial proteomics, where permutation-based neighborhood enrichment / co-occurrence analyses are widely applied in standard workflows. Our comparison therefore provides practical guidance, especially for users with limited computational background, to make informed method choices.

ACTION: In the revised manuscript we added an additional part to the introduction to further articulate the motivation and relevance of a method comparison early in the manuscript.

“Tissues show subtle, heterogeneous spatial differences that can lead to markedly different results depending on the NEP method used. One key difference between methods is how they capture NEP directionality.” (lines 69-71)

“Motivated by this fundamental conceptual difference, this study explores the landscape of commonly used NEP methods. We evaluate their performance on simulated and real-world datasets ranging from the detection of clear cohort differences to the identification of more subtle and directional NEPs.” (lines 83 - 86)

We also extended the discussion to highlight both the variability of the methods and the tissue structures that can be recovered:

“This work not only highlights the heterogeneity of the available methods and their results, but also the wide range of subtle and heterogeneous spatial differences that can be recovered from spatial data. With an appropriate score, NEP methods can recover the strength of neighbor preferences, which is often the biologically relevant quantity rather than mere detectability. We also revealed that directionality is biologically meaningful yet inconsistently captured across existing approaches.” (lines 683-689)

(2). The new method proposed here, COZI, does not appear to outperform existing methods.

ANSWER: We thank reviewer 1 for this comment. COZI was not designed to maximize performance on a single comparison parameter. Throughout our systematic comparison, we identified (1) sensitivity (Figure 2) and (2) directionality (Figure 3-4) as major differentiation axes between NEP methods. Only COZI enables both sensitivity and directionality while remaining robust across various data characteristics (e.g., imbalanced cell-type abundances and subtle spatial differences). While performance is comparable to (or better than) existing approaches, COZI uniquely offers additional biological insights that other methods cannot recover (Figure 5-6). By introducing the conditional cell ratio, we specifically focus on the biological interpretability of COZI.

ACTION: In the revised manuscript, we adapted the abstract to highlight the uniqueness and benefits of COZI.

“Overall, our study serves as a comprehensive guide for users and method developers in spatial omics analysis and offers a novel approach (COZI), which enables both sensitivity and directionality while remaining robust across various data characteristics, to performing NEP analysis.”(lines 35-38)

We further clarify both the uniqueness and limitations of COZI in the discussion.

“Overall, throughout our systematic comparison, we identified (1) sensitivity and (2) directionality as major differentiation axes between NEP methods. Only COZI enables both while remaining robust across various data characteristics (e.g., imbalanced cell-type abundances and subtle spatial differences). While performance is comparable to (or better than) existing approaches, COZI uniquely offers additional biological insights that other methods cannot recover.

A limitation when using COZI is that interpretation based solely on z-scores provides an incomplete view, especially in more complex tissues. Only the CCR, in combination with z-scores, enables a fine-grained interpretation of spatial images.” (lines 708-716)

Reviewer #2 (Remarks to the Author):

The authors have responded effectively to my major points. The addition to the discussion addressing the challenge of spot-based technologies is welcome. It should be noted that not all these technologies could be described as ‘lower’ resolution (e.g. Stereo-Seq or Visium HD have individual bins that could provide subcellular resolution), but an additional step would be required to approach data similar to the single-cell data used previously (e.g. the use of software such as bin2cell to estimate cell boundaries).

Additionally, I appreciate the greater clarity the authors have provided for the mouse data used in the study. The additional description provided makes it easier for the reader to understand the exact data used.

Finally, the comprehensive response to the other reviewers’ points have improved the manuscript – particularly the additional analysis provided in figure 4 using SpaSim data simulations, which provides welcome additional evidence of the effectiveness of COZI.

I’m not sure the clarification added to the manuscript with regards to single-sample vs multi-sample comparisons is very helpful to the uninitiated reader (lines 201-203). While I understand the authors’ perspective regarding applications in basic vs translational biology, I think this addition could certainly be strengthened.

ANSWER: We thank reviewer 2 for the positive feedback and are pleased that the revisions have improved the clarity and strength of the manuscript. We appreciate the comment regarding the explanation of single-sample versus multi-sample comparisons and have further revised this section to improve accessibility for readers who may be unfamiliar with the concept.

ACTION: In the revised paragraph, we provide better description and a concrete example to aid comprehension. We believe the revised version strengthens the explanation and improves overall readability.

“NEP analysis generates one preference score per image for each pair of cell types. This allows comparisons such as whether the purple cell type exhibits a stronger preference to neighbor the red cell type in one image than in another. When multiple samples are available per cohort, these per-image NEP scores can then be compared across samples to detect cohort-level differences, a common practice in the field. Our first experimental setup is built on this common practice: If a distinct NEP signature exists in one cohort but not the other, the methods should detect this difference.” (lines 208-214)

Reviewer #2 (Remarks on code availability):

I have no additional comments on the code at this stage.

Reviewer #3 (Remarks to the Author):

For the most part, the authors have addressed reviewers' comments well and thoroughly. Below are followups to my comments:

1. It's fair to say that unrealistic assumptions are required to derive closed form expressions of the mean and variance of COZI. However, is permutation really a less unrealistic assumption? Randomizing the cell type labels also assumes that labels are randomly distributed independently of tissue structure as the null hypothesis.

ANSWER: We thank reviewer 3 for this important and topical question. While permutations are an unbiased estimate of a null that do not require parametric assumptions about spatial independence, we acknowledge that global permutations are still a simplistic null.

In our work, we focused on a comparison of already existing NEP methods to guide the user's method choice. These mainly include permutation based methods but also methods with other null hypothesis assumptions like Cellcharter where we report that permutation based methods prove to perform best.

There are other studies exploring alternative space restricted null hypotheses that we now include in the discussion as future direction. The recent work by Ding et al. (2025, <https://doi.org/10.1038/s41588-025-02119-z>) identifies hotspots of high diversity in an image to restrict colocalization analysis to the hotspot patches. The recent work by Ameen et al. (2025, <https://doi.org/10.1016/j.crmeth.2025.101175>) describes the method *Kontextual* to introduce the idea of performing random permutations in a subcontext of the whole image, e.g. CD8 cell permutation only across all immune and not all cells to retain tissue structure.

ACTION: Based on your comment, we now included a discussion on the limitation of permutations and further describe the idea of a context dependent null hypothesis with a reference to the mentioned studies above.

“Another important discussion point are assumptions regarding tissue structure when performing whole-slide permutation of cell type labels, as used in COZI and many other NEP methods. Whole slide spatial randomization assumes that cell types are randomly distributed independently of tissue structure, an assumption that is unrealistic in many biological contexts. A key example is the presence of large tumor cell clusters, which naturally form contiguous aggregates rather than random distributions. With this thought, the recently introduced Kontextual framework by Ameen et al. (2025) provides a different conceptual idea by performing context-restricted permutations. They propose to randomize “child” cell types only across their “parent” cell type areas, for example permuting CD8⁺ cells only within their parent immune cell compartment rather than across the entire image. In Ding et al. (2025) they first identify hotspots of high diversity in tissues and only afterwards perform colocalization analysis in these hotspot patches. Both methods thereby preserve aspects of tissue architecture while still enabling statistical testing. While the methods require clear biological hypotheses, it is one direction of interest for more realistic NEP inference in the future.” (lines 725-740)

2. While correlations shown in Supplementary Figure 11 are generally high, some of them are not very high. Some of the spatial neighborhood graphs may have different biological meanings. For example, triangulation, knn with small k, and distance-based neighbors with a

small radius give closer neighborhoods, while knn with larger k and distance-based neighbors with a larger radius include more distant neighbors. When the different graphs give different results, would the different results give interesting different aspects of biology? It might be interesting to illustrate this on the myocardial infarction or the breast cancer example, with a graph with only close neighbors and another with some more distant neighbors.

ANSWER: This is a very interesting question and therefore we now included a new Supplementary Figure 12 with myocardial immune cell infiltration plots similar as in Figure 6 c and f ($k = 5$), but now choosing small neighborhoods (radius 50 pixels) as well as larger neighborhoods (knn with $k = 15$ and 30). As suggested, this captures the extremes of small and large neighborhoods on one of our biological dataset.

The results show preserved NEP scores across all three neighborhood definitions. Importantly, the conditional cell ratio as well as the range of z-scores increases with increasing neighborhood size as the number of interactions counted per index cell increases drastically. The results align with our findings from Figure 6 showing infarct core infiltration of neutrophils before monocyte infiltration (Supplementary Figure 12 a-c). Also the monocyte NEP for Endocardial cells at 24h is recovered across all neighborhood definitions (Supplementary Figure 12 d-f).

This indicates that the recovered NEPs are already apparent with close neighbors and either persist or do not get hidden by including more distant neighbors into the neighborhood definition.

ACTION: We included a new Supplementary Figure 12 highlighting the robustness of neighborhood choice in the revised manuscript (line 665-677, line 1117-1122).

“We further explored myocardial immune cell infiltration plots similar as in Fig. 6 c and f ($k=5$), but now choosing small neighborhoods (radius 50 pixels) as well as larger neighborhoods (knn with $k = 15$ and 30) (Supplementary Fig. 12). The results show preserved NEP scores across all three neighborhood definitions. Importantly, the conditional cell ratio as well as the range of z-scores increases with increasing neighborhood size as the number of interactions counted per index cell increases drastically. The results align with our findings from Fig. 6 showing infarct core infiltration of neutrophils before monocyte infiltration (Supplementary Fig. 12 a-c). Also the monocyte NEP for Endocardial cells at 24h is recovered across all neighborhood definitions (Supplementary Fig. 12 d-f).

This indicates that the recovered NEPs are already apparent with close neighbors and either persist or do not get hidden by including more distant neighbors into the neighborhood definition.” (lines 665-677)

Supplementary Figure 12: COZI NEP findings across neighborhood scales on the MI dataset. COZI results on the MI dataset with varying neighborhood definitions to compare to Figure 6 c and f (kNN with k=5). COZI was calculated with different neighborhood definitions for small neighborhoods (fixed radius: 50) and larger neighborhoods (kNN with k = 15, 30). COZI z-scores (colorscale) with conditional cell ratio (dotsize) for the three neighborhood definitions for (a-c) monocyte and neutrophil NEP with stressed cardiomyocytes and (d-f) monocyte and neutrophil NEP with endocardial cells.

3. In the caption of Supplementary Figure 6: "4h and 24h after infiltration" -- did you mean "infarction"?

ANSWER: Yes, thank you.

ACTION: We corrected the caption in the revised manuscript (Supplementary Figures, line 39)

Reviewer #3 (Remarks on code availability):

I still think having a stand-alone package for COZI is better than the pull requests, so it's more modular. Users who want to use COZI but not other functionalities of the heavier packages where you submitted pull requests don't have to install the functionalities they don't need. For example, more people analyzing spatial transcriptomics data use Seurat rather than imcRtools.

ANSWER: We agree with reviewer 3 that providing COZI in both a standalone package and common analysis toolboxes is best for allowing the community the most flexible usage of the method. Therefore, we now provide the package COZipy (<https://pypi.org/project/cozipy/>) that allows users to run COZI with the three neighborhood definitions with minimal

dependencies. The user only has to provide x and y-coordinates and cell type labels. As we still understand COZI as a flavor of NEP analysis, we are happy to share that a conditional z-score (COZI) is now officially part of IMCRtools (primarily spatial proteomics community) and will be part of Squidpy (primarily spatial transcriptomics community) soon.

ACTION: We implemented the python package COZipy (<https://pypi.org/project/cozipy/>) that is now available for the community for flexible usage of the method.

“COZI can be used in the python package COZipy (<https://pypi.org/project/cozipy/>) as well as in IMCRtools (when choosing the “conditional” method)” (lines 1169-1171)

REVIEWERS' COMMENTS

Reviewer #1 (Remarks to the Author)

This revision does not present any new evidence to further support the significance of NEP benchmark analysis or to demonstrate the additional power of COZI compared to existing methods. Therefore, my original criticisms still stand.

We strongly believe that our previous responses and manuscript revisions adequately address this concern within the intended scope of the study. Nevertheless, in light of this comment, we have further addressed our motivation and rationale in the introduction (lines 74–77). We also revised the introduction (lines 91–92), results (lines 403–405), and discussion (lines 626–634).

(Remarks on code availability)

Reviewer #3 (Remarks to the Author)

The authors have satisfactorily addressed remaining comments regarding the manuscript.

We thank Reviewer 3 for their constructive comments throughout the revision process. We greatly appreciate the reviewer's input, which helped to further improve the manuscript.

(Remarks on code availability)

While having the pip package is nice, I don't think having COZI as part of imcRtools is a good idea. While COZI is of great interest to spatial transcriptomics, imcRtools is rarely used in transcriptomics. Having a stand alone R package can increase the visibility of COZI.

We thank Reviewer 3 for emphasizing the expected great interest in COZI and have therefore implemented `coziR` (<https://github.com/SchapiroLabor/coziR>). With this, spatial proteomics and transcriptomics researchers can run COZI in both R and Python.